# AudioMarkBench: Benchmarking Robustness of Audio Watermarking

**Hongbin Liu**[*1], **Moyang Guo**[*1], **Zhengyuan Jiang**[1], **Lun Wang**[2], **Neil Zhenqiang Gong**[1]

[1]Duke University, [2]Google

[1]{hongbin.liu, moyang.guo, zhengyuan.jiang, neil.gong}@duke.edu, [2]lunwang@google.com

## Abstract

The increasing realism of synthetic speech, driven by advancements in text-to-speech models, raises ethical concerns regarding impersonation and disinformation. Audio watermarking offers a promising solution via embedding human-imperceptible watermarks into AI-generated audios. However, the robustness of audio watermarking against common/adversarial perturbations remains understudied. We present AudioMarkBench, the first systematic benchmark for evaluating the robustness of audio watermarking against *watermark removal* and *watermark forgery*. AudioMarkBench includes a new dataset created from Common-Voice across languages, biological sexes, and ages, 3 state-of-the-art watermarking methods, and 15 types of perturbations. We benchmark the robustness of these methods against the perturbations in no-box, black-box, and white-box settings. Our findings highlight the vulnerabilities of current watermarking techniques and emphasize the need for more robust and fair audio watermarking solutions. Our dataset and code are publicly available at `https://github.com/moyangkuo/AudioMarkBench`.

## 1 Introduction

Recent advancements in text-to-speech (TTS) generative models enable generating highly realistic synthetic audios that are indistinguishable from real human voices. However, this capability raises significant concerns, such as malicious impersonation, dissemination of false information, or copyright infringement. For example, a scammer used synthetic audios to impersonate President Biden in illegal robocalls during a New Hampshire primary election, and thus faces a $6 million fine and felony charges [5].

Audio watermarking [14, 3, 12] offers a promising approach to mitigate concerns about synthetic audio authenticity. It embeds an imperceptible watermark into a synthetic audio using a watermark encoder, outputting a watermarked audio. During detection, a watermark decoder extracts a watermark from a given audio input. By comparing the extracted watermark with the ground-truth watermark, one can determine the authenticity of the given audio.

Existing audio watermarking methods perform well when there are no perturbations added to watermarked audios [14, 3, 12]. However, real-world audios often undergo various perturbations. Common perturbations include compression using standards like MP3 or Opus [15] to reduce internet transmission costs. Additionally, attackers may craft adversarial perturbations designed to deceive watermarking methods. However, the robustness of audio watermarking against these perturbations remains under-explored and lacks systematic benchmarking.

**Our work:** In this work, we aim to bridge the gap by introducing **AudioMarkBench** (**Audio** Water**mark**ing **Bench**mark), the *first* systematic and comprehensive benchmark for assessing the

---

[*]Equal contributions.

38th Conference on Neural Information Processing Systems (NeurIPS 2024) Track on Datasets and Benchmarks.

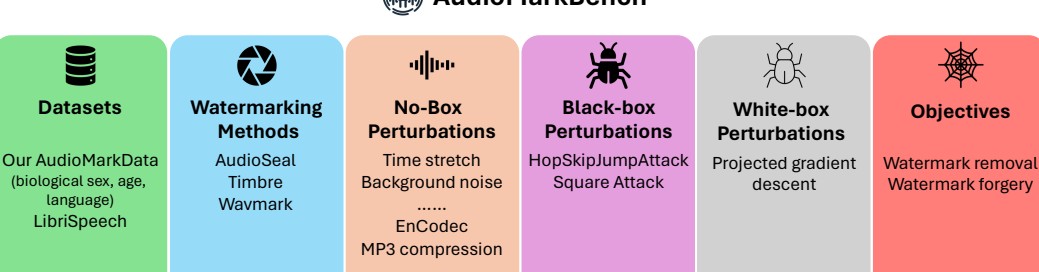

Figure 1: Summary of our AudioMarkBench.

robustness of audio watermarking. We focus on evaluating robustness against two types of perturbations: *watermark-removal* perturbations, designed to make watermarked audio undetectable, and *watermark-forgery* perturbations, which aim to falsely mark unwatermarked audio.

**-  *Datasets*:** Other than the standard LibriSpeech dataset [13], we construct a new dataset AudioMark-Data that meticulously sub-samples 20,000 audio samples from the Common Voice dataset [2], striving to ensure balanced representation of biological sexes, languages, and age groups. Moreover, our datasets provide not only watermarked/unwatermarked audios but also perturbed audios under various perturbations, making it easier for future research to assess the the effectiveness of new watermark-removal/forgery perturbations.

**-  *Systematic benchmarking*:** We present the *first* systematic benchmark evaluating the robustness of three state-of-the-art audio watermarking methods against 15 different watermark-removal/forgery perturbations across two datasets. Twelve of these perturbations, termed "no-box" perturbations, require no access to the watermarking method. These perturbations include common audio edits like codec [7, 16, 15] and audio filter, and noise addition such as white noise or background noise. Additionally, We adapt two adversarial example methods [4, 1] in the *black-box* setting (*i.e.*, access to watermark detector API only) and one adversarial example method [10] in the *white-box* setting (*i.e.*, full access to watermarking model parameters) from image classifiers to audio watermarking.

**-  *Findings*:** We make intriguing findings in our benchmark study. First, we confirm that all studied audio watermarking methods can distinguish watermarked/AI-generated audios from unwatermarked/non-AI-generated audios precisely when no perturbations are added. Second, existing audio watermarking methods can be vulnerable to watermark removal including certain no-box perturbations (e.g., EnCodeC [7]), black-box perturbations with sufficient quota for API queries, and white-box perturbations. Third, current audio watermarking techniques are effective at resisting no-box and black-box watermark forgery, but vulnerable to white-box forgery. Fourth, existing audio watermarking methods have robustness gaps among biological sex groups (female vs male) and language groups under certain perturbations, flagging potential fairness issues. However, we do not observe consistently significant robustness gaps across age groups.

## 2   Audio Watermarking Methods

An audio watermarking method consists of four key components: a watermark $w$, encoder `Enc`, decoder `Dec`, and detector `Det`. The watermark $w \in \{0, 1\}^n$ is typically an $n$-bit bitstring, such as an 16-bit bitstring 1110110110010110. Given any audio waveform $s \in \mathbb{R}^T$ and a watermark $w$, the encoder `Enc` outputs a watermarked audio waveform $s_w = \text{Enc}(w, s) \in \mathbb{R}^T$, where $T$ denotes the number of time samples in the waveform. For any audio waveform $s$, whether watermarked or unwatermarked, the decoder `Dec` can extract a bitstring watermark $\text{Dec}(s)$. When the audio waveform $s$ is watermarked with $w$, the extracted watermark $\text{Dec}(s)$ should be similar to $w$. The detector `Det` then uses the decoded watermark $\text{Dec}(s)$, together with some additional information, to determine if the given audio waveform $s$ contains a watermark. In particular, $\text{Det}(s) = 1$ ($\text{Det}(s) = 0$) means that $s$ is detected as watermarked (unwatermarked), respectively. In this study, we examine three state-of-the-art, open-source audio watermarking techniques: AudioSeal/AudioSeal-B [14],

Timbre [12], and WavMark [3]. We utilize the publicly available code and models of these methods for our experimental analysis.

**AudioSeal/AudioSeal-B:** During watermark generation, AudioSeal uses a sequence-to-sequence encoder `Enc` to generate the watermarked waveform $s_w$ given any input audio waveform $s$ and a watermark $w$. During watermark detection, the decoder `Dec` gives two outputs given a suspect waveform $s$: a global detection probability $P_s$ indicating the likelihood that $s$ is watermarked, and the decoded watermark `Dec`$(s)$. With a detection threshold $\tau$, AudioSeal predicts `Det`$(s) = 1$ if the detection probability $P_s$ exceeds $\tau$, and 0 otherwise. AudioSeal-B, a variant of AudioSeal, uses *bitwise accuracy* (*i.e.* the proportion of matching bits between two bitstrings) for detection instead. Specifically, it predicts `Det`$(s) = 1$ if the bitwise accuracy between the decoded watermark and the original watermark is at least $\tau$: `BA`(`Dec`$(s), w) \geq \tau$, and 0 otherwise.

**Timbre:** Given any input audio $s$, Timbre first transforms it into a spectrogram $C_s = (a_s, p_s)$ using Short-Time Fourier Transformation (STFT), where $a_s$ is the amplitude and $p_s$ is the phase. It then embeds the watermark $w$ into $a_s$ while keeping $p_s$ unchanged, producing the watermarked audio $s_w = $ `ISTFT`(`Enc`$(a_s, w), p_s)$, where ISTFT is the inverse STFT. For detection, given an audio waveform $s$, STFT is first applied to obtain its spectrogram $C_s = (a_s, p_s)$, and then the decoder `Dec` extracts a watermark `Dec`$(a_s)$ from the amplitude $a_s$. The detector outputs `Det`$(s) = 1$ if the bitwise accuracy `BA`(`Dec`$(a_s), w) \geq \tau$, otherwise `Det`$(s) = 0$.

**WavMark:** Similar to Timbre, WavMark operates in the spectrogram domain by first transforming an input waveform $s$ to its spectrogram $C_s = (a_s, p_s)$ via STFT. It then embeds a preset synchronization bitstring $s_{\text{sync}}$ together with the watermark $w$ into the whole spectrogram, i.e., producing the watermarked audio $s_w = $ `ISTFT`(`Enc`$(C_s, s_{\text{sync}} \cup w))$ where $\cup$ denotes bitstring concatenation. For detection, given an audio waveform $s$, the decoder extracts a bitstring containing both a decoded synchronization bitstring `Dec`(`STFT`$(s))_{\text{sync}}$ and watermark `Dec`(`STFT`$(s))_w$ from its spectrogram. If the decoded synchronization bitstring `Dec`(`STFT`$(s))_{\text{sync}} = s_{\text{sync}}$ and the bitwise accuracy `BA`(`Dec`(`STFT`$(s)), w) \geq \tau$, then `Det`$(s) = 1$, otherwise `Det`$(s) = 0$.

**Importance of determining a detection threshold $\tau$:** In real-world deployments, the detector determines whether an audio waveform $s$ contains a watermark or not by comparing metrics, such bitwise accuracy, with the detection threshold $\tau$. Thus, $\tau$ controls a trade-off between *False Positive Rate (FPR)* and *False Negative Rate (FNR)*, where FPR (or FNR) is the likelihood of incorrectly predicting an unwatermarked (or watermarked) audio as watermarked (or unwatermarked). A higher $\tau$ reduces FPR but increases FNR. $\tau$ can vary depending on the specific watermarking method and we will further discuss selecting $\tau$ in our experiments in Section 5.

## 3   Watermark-removal and Watermark-forgery Perturbations

**Definitions:** Audio watermarking faces two primary threats: *watermark-removal perturbations*, which aim to strip watermarks from watermarked audios, and *watermark-forgery perturbations*, which aim to forge watermarks for unwatermarked audios. Watermark removal allows AI-generated audio to be falsely presented as genuine, potentially fueling disinformation campaigns. Conversely, watermark forgery can mislabel authentic audio as AI-generated, undermining human creators' ability to claim ownership and potentially stifling human creativity.

**-  *Watermark removal*:** Watermark removal aims to add a human-imperceptible perturbation vector $\delta$ to a watermarked audio $s_w$ such that the detector `Det` outputs 0 for $s_w + \delta$. Formally, finding $\delta$ can be formulated as the following optimization problem:

$$\delta_{\text{removal}} = \arg\min_\delta \quad \text{Det}(s_w + \delta) = 0 \quad \text{s.t.} \quad Q(s_w + \delta) \approx Q(s_w), \tag{1}$$

where $Q$ is an audio quality metric. The quality constraint ensures the audio quality to remain high after adding the perturbation. The audio quality metric $Q$ can be ViSQOL [9] or SNR.

**-  *Watermark forgery*:** In contrast, watermark forgery attempts to add perturbation $\delta$ to an unwatermarked audio $s_u$ such that the detector `Det` detects it as watermarked. Formally, finding $\delta$ in watermark forgery can be formulated as the following optimization problem:

$$\delta_{\text{forgery}} = \arg\min_\delta \quad \text{Det}(s_u + \delta) = 1 \quad \text{s.t.} \quad Q(s_u + \delta) \approx Q(s_u). \tag{2}$$

Both watermark removal and watermark forgery perturbations can be classified into three groups based on the adversary's knowledge of the watermarking method.

**No-box perturbations:** In no-box setting, the perturbations are crafted without any knowledge of the audio watermarking method, including the architecture, parameters, or even the output of the detector. These no-box perturbations are created blindly or even unintentionally to spoof the watermarking detector for watermark removal/forgery. In our AudioMarkBench, we consider twelve common audio editing operations as no-box perturbations, including Gaussian/background noises and audio codecs like MP3, EnCodeC, SoundStream, Opus, *etc*. More details on these perturbations can be found in Appendix A.2.

**Black-box perturbations:** In black-box setting, perturbations are created by interacting with the watermarking detector `Det` as an oracle. Specifically, the attacker can choose audios to submit to the detector and observe the detection result without any knowledge of how the detector operates internally. We extend existing methods for finding black-box adversarial examples [4, 1] against image classifiers to audio watermarking detectors. In particular, we apply them in waveform and/or spectrogram domains. Next, we briefly describe how we extend them, and Appendix A.3 shows more technical details.

- *HopSkipJumpAttack (HSJA) [4]:* Given an audio $s$ and access to a watermarking detector `Det`'s output, HopSkipJumpAttack iteratively approximates `Det`'s decision boundary to find a minimal watermark-removal/watermark-forgery perturbation $\delta$. We implement this attack in both waveform and spectrogram domains. In the waveform domain, perturbations are optimized in a 1-D vector space, while in the spectrogram domain, both phase and amplitude (2-D vectors) are optimized. We conduct 10,000 iterations in each domain, initializing perturbations with Gaussian noise.

- *Square attack [1]:* Given an audio $s$ and access to a watermarking decoder `Dec`'s output, Square attack iteratively finds the watermark-removal/watermark-forgery perturbation $\delta$ by strategically decreasing either the bitwise accuracy of `Dec`'s output or the global detection probability (for AudioSeal). We extend Square attack from image domain to the spectrogram domain by treating a spectrogram as an image. Note that Square attack is only applicable to the spectrogram domain (not the waveform domain) since its input is a 2-D image/spectrogram. We perform Square attack under a $\ell_\infty$-norm perturbation constraint for 10,000 iterations.

**White-box perturbations:** In white-box setting, the perturbations are crafted with full knowledge of the watermark decoder `Dec`'s parameters and the ground-truth watermark $w$. In particular, the perturbations are found via solving the optimization problems in Equation 1 and 2. The goal of watermark forgery/removal perturbation is to increase/decrease the bitwise accuracy between the decoded watermark `Dec`$(s)$ and ground-truth watermark $w$ (for Timbre, WavMark, and AudioSeal-B) or the global detection probability (for AudioSeal). Therefore, for Timbre, WavMark, and AudioSeal-B, we use the cross-entropy loss to minimize/maximize the distance between the decoded watermark `Dec`$(s + \delta)$ and ground-truth watermark $w$:

$$L_{ce} = -\sum_{i=1}^{n} w_i \log(\text{Dec}(s + \delta)_i) + (1 - w_i) \log(1 - \text{Dec}(s + \delta)_i)$$

, where $w_i$ (or `Dec`$(s + \delta)_i$) is the $i^{th}$ bit of $w$ (or `Dec`$(s + \delta)$). For AudioSeal, we adopt the ReLU activation applied between the global detection probability $P_s$ and $\tau$: $L_{Re} = \max(0, P_s - \tau)$. We use these loss functions to approximate the objective functions in Equation 1 and 2. Appendix A.4 shows more details on how we solve the optimization problems to find white-box perturbations.

## 4 Datasets

**Unwatermarked audio samples:** Our AudioMarkBench includes two datasets of unwatermarked audio samples, i.e., AudioMarkData and LibriSpeech [13]. AudioMarkData is a dataset we build from the Common Voice dataset [2]. Each audio sample in AudioMarkData is associated with three attributes, which are: *language* (25 languages), *biological sex* (male, female) and *age* (teens, twenties, thirties, fourties). We use these attributes to benchmark whether watermarking methods have different performance/robustness for audio samples with different attributes. For every attribute group (language, biological sex, age), AudioMarkData samples 100 audio samples in 5 seconds with sampling rate at 16kHz from Common Voice, resulting in 20,000 audio samples in total. Table 1

Table 1: Attributes of AudioMarkData. Details of languages are shown in Appendix A.1.

| Attribute | #Values | Values | #Samples per Value |
|---|---|---|---|
| Language | 25 | EU, BE, BN, YUE, CA, ZH-CN, ZH-HK, ZH-TW, EN, EO, FR, KA, DE, HU, IT, JA, LV, MHR, FA, RU, SW, ES, TA, TH, UK | 800 |
| Biological Sex | 2 | Male, Female | 10,000 |
| Age | 4 | Teens, Twenties, Thirties, Forties | 5,000 |

summarizes the attributes of AudioMarkData. The LibriSpeech dataset contains over 1,000 hours of read English speech derived from audiobooks in the public domain. We sampled 20,000 audio samples with a maximum length of 5 seconds at the default 16kHz sampling rate. Note that audio samples in LibriSpeech do not have attributes.

**Watermarked audio samples:** We apply each watermarking method (AudioSeal/AudioSeal-B, Timbre, and WavMark) to embed a watermark into each audio sample. Note that AudioSeal/AudioSeal-B use the same encoder and decoder, but different detectors. Specifically, we randomly sample a 16-bit watermark for each watermarking method and embed it into each audio sample. In total, we create 20,000 watermarked audio samples for each watermarking method and each dataset.

**Perturbed audio samples:** We add watermark-removal (or watermark-forgery) perturbations to watermarked (or unwatermarked) audio samples to create perturbed audio samples. These perturbed audio samples will be used to measure the robustness of audio watermarking against watermark removal/forgery. Specifically, we consider 12 categories of common no-box perturbations. For each category of no-box perturbation, we utilize it to perturb the 20,000 unwatermarked audio samples in each dataset and the 20,000 watermarked audio samples in each dataset and watermarking method. Note that each category of no-box perturbations has certain parameter to control the level of perturbation, and we use multiple parameter values (see Appendix A.2). For the black-box and white-box perturbations, due to limits of computation resources, we sample 200 unwatermarked audio samples and 200 watermarked audio samples for each watermarking method in the LibriSpeech dataset; and in AudioMarkData, we sample one unwatermarked audio sample and one watermarked audio sample from each attribute group (language, biological sex, age), leading to 200 unwatermarked audio samples and 200 watermarked audio samples for each watermarking method.

## 5 Benchmark Results

In the following section, we present our primary benchmark results and findings. We conduct our experiments on 18 NVIDIA-RTX-6000 GPUs, each with 24 GB memory. The complete set of experiments requires about 430 GPU-hours to execute.

### 5.1 Evaluation Metrics

We use FNR and FPR to evaluate the robustness of audio watermarking. Specifically, FNR/FPR is the fraction of watermarked/unwatermarked audios that are incorrectly detected as unwatermarked/watermarked. Lower FNR/FPR indicate better audio watermarking methods. When watermarked audios (or unwatermarked audios) are modified by watermark-removal (or watermark-forgery) perturbations, lower FNR (or FPR) indicates that the watermarking method is more robust against watermark removal (or watermark forgery).

We evaluate the quality of perturbed audios using standard metrics including SNR and ViSQOL [9]. Signal-to-Noise Ratio (SNR) evaluates quality of a perturbed audio by comparing its level of noise with the corresponding clean audio (called *reference audio*), where the reference audio is watermarked (or unwatermarked) in watermark removal (or forgery). Higher SNRs indicate clearer and higher-quality perturbed audios. ViSQOL, ranging from 1 to 5, evaluates audio quality by simulating human perception of audios, where a higher score indicates the perturbed audio better preserves quality of the reference audio. A ViSQOL score no smaller than 3 generally reflects good audio quality. We mainly rely on ViSQOL for measuring audio quality because it is more reliable than SNR [9].

### 5.2 Results under No Perturbations

Figure 2 and Figure 9 (in Appendix) show the FPR and FNR of each watermarking method as the detection threshold $\tau$ varies on AudioMarkData and LibriSpeech datasets, respectively. No

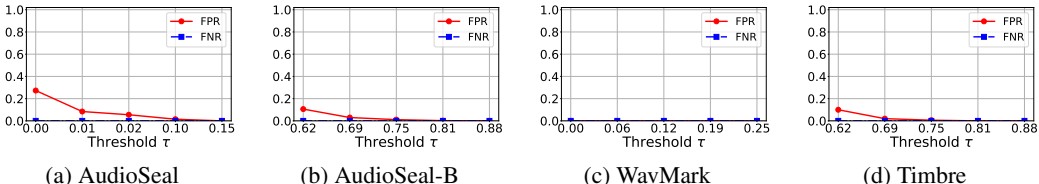

|(a) AudioSeal|(b) AudioSeal-B|(c) WavMark|(d) Timbre|

Figure 2: Detection results under no perturbations on AudioMarkData. We set the detection threshold $\tau$ for each watermarking method as follows: AudioSeal $\tau = 0.15$, AudioSeal-B $\tau = 0.875$, WavMark $\tau = 0.0$, and Timbre $\tau = 0.8125$, to achieve FPR $< 0.01$ and FNR $< 0.01$. Results for LibriSpeech are in Figure 9 in Appendix.

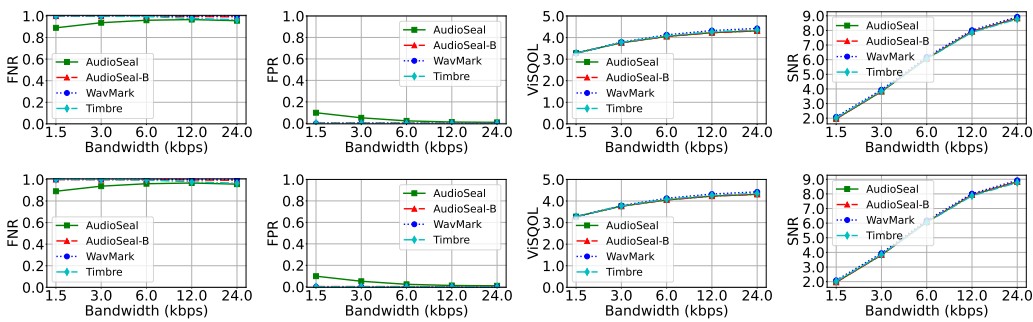

Figure 3: Detection results under EnCodeC perturbations on both datasets (first row: AudioMarkData and second row: LibriSpeech). Results of the other eleven no-box perturbations are in Appendix A.6.

perturbations are added to watermarked/unwatermarked audios. We have three key observations. First, FNRs of each watermarking method on both datasets are close to 0 for a wide range of detection threshold $\tau$, indicating that watermarked audios can be accurately detected as watermarked. Second, FPRs of each watermarking method on both datasets decrease as detection threshold $\tau$ increases. This is because unwatermarked audios are less likely to be falsely detected as watermarked when $\tau$ increases. Third, audio watermarking methods are very accurate at distinguishing watermarked and unwatermarked audios when the detection threshold $\tau$ is properly selected. For instance, when $\tau = 0.15$, both FPR and FNR of AudioSeal are almost 0 on AudioMarkData. For each watermarking method, we choose the smallest detection threshold $\tau$ that achieves both FNR and FPR lower than 0.01. The selected $\tau$ for each watermarking method and each dataset is shown in the captions of Figure 2 and Figure 9. In the rest of this paper, we will use these detection threshold $\tau$ unless otherwise mentioned.

### 5.3 Robustness against No-box Perturbations

Figure 3 shows the FNR, FPR, SNR, and ViSQOL results of the watermarking methods against EnCodeC perturbations on AudioMarkData and LibriSpeech datasets. Results of the other eleven no-box perturbations can be found in Appendix A.6.

**Overall results:** We have several key observations. First, state-of-the-art audio watermarks are robust against several common no-box watermark-removal perturbations such as time stretch, low-pass, high-pass, and echo. Specifically, while preserving the quality of watermarked audio samples well (i.e., ViSQOL no smaller than 3), those perturbations have small impact on FNRs. This is because these audio watermarking methods use *adversarial training* [8], which considers various common no-box perturbations, to train the encoders and decoders. Second, current audio watermarking methods are not robust against no-box removal perturbations that are unseen during adversarial training. For instance, when ViSQOL is no smaller than 3, EnCodeC, SoundStream, and Opus achieve very high FNRs, indicating that those perturbations can remove watermarks from watermarked audios while preserving the audio quality. Third, current audio watermarking methods have good robustness against watermark-forgery perturbations. In particular, FPRs of all these watermarking methods are almost always close to 0, except for quantization. Specifically, when bit levels are smaller than 32,

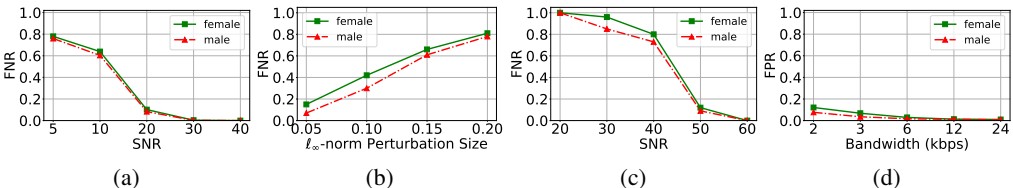

(a)            (b)            (c)            (d)

Figure 4: FNRs in biological sexes against watermark-removal (a) Gaussian noise perturbations, (b) Square attack perturbations, and (c) white-box perturbations. (d) FPRs in biological sexes against watermark-forgery EnCodeC perturbations. The watermarking method is AudioSeal. The gaps between "female" and "male" are statistically significant in two-tailed t-test with $p$-value $< \alpha = 0.05$.

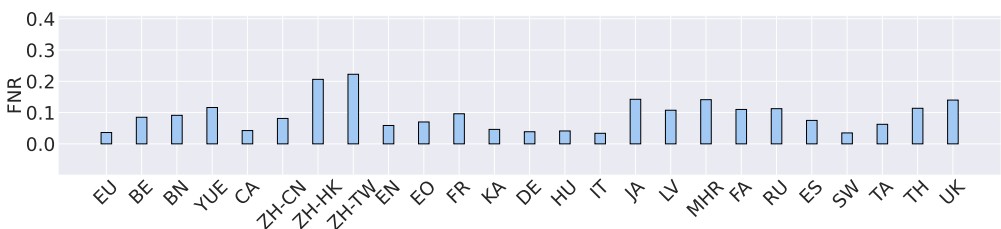

Figure 5: Language difference against watermark-removal Gaussian noise perturbations with SNR 20. The watermarking method is AudioSeal.

quantization perturbation achieves a FPR larger than 0.2, but the audio quality is also compromised. This is because forging a watermark is harder and may require knowledge of the watermarking model. No-box perturbations do not have such information and therefore cannot forge a watermark. As we will show in the next subsection, forging a watermark remains difficult even in the black-box setting.

**Comparing watermarking methods:** Considering performance against all no-box perturbations, AudioSeal is the most robust against watermark removal and forgery among the evaluated watermarking methods. In contrast, WavMark is the least robust. For instance, watermarks embedded by WavMark can even be removed by Gaussian noise and MP3 compression without compromising the watermarked audios' quality. This stems from two reasons: 1) AudioSeal uses advanced sequence-to-sequence models as encoder and decoder, which can output fine-grained localization of watermarks; and 2) AudioSeal considers more diverse perturbations in adversarial training.

**Comparing biological sex, language, and age groups in AudioMarkData:** Figure 4 and more results in Appendix A.7 show detection differences in biological sexes in terms of FNRs/FPRs. First, watermarked audios with attribute "female" are less robust to watermark-removal Gaussian noise perturbations (i.e., have higher FNRs) than those with attribute "male" for all the evaluated watermarking methods especially AudioSeal-B. These results indicate a fairness gap of robustness against watermark removal among "female" and "male" groups under Gaussian noise perturbations. To rigorously test this gap, we conduct a two-tailed t-test with a null hypothesis positing no difference in FNRs between "female" and "male" groups, at a significance level of $\alpha = 0.05$. For Figure 4a, the calculated $p$-value$\approx 2.4 \times 10^{-6} < \alpha = 0.05$. Thus, the robustness gap between "female" and "male" groups is statistically significant. Note that we did not observe such gaps for other watermark-removal no-box perturbations except EnCodeC (Figure 18), Opus (Figure 19), Quantization (Figure 20).

Second, unwatermarked audios with attribute "female" are less robust (i.e., have larger FPRs) to watermark-forgery EnCodec perturbations than those with attribute "male" when AudioSeal is used. We did not observe such gaps for other watermarking methods under EnCodec perturbations nor other watermark-forgery no-box perturbations for all watermarking methods since FPRs are generally close to 0 in those scenarios.

Figure 5 and results in Appendix A.9 show detection differences in languages in terms of FNRs/FPRs. We observe noticeable differences across languages. In particular, watermarked audios in Georgian have relatively smaller FNRs against Gaussian noise, Background noise, and Quantization perturba-

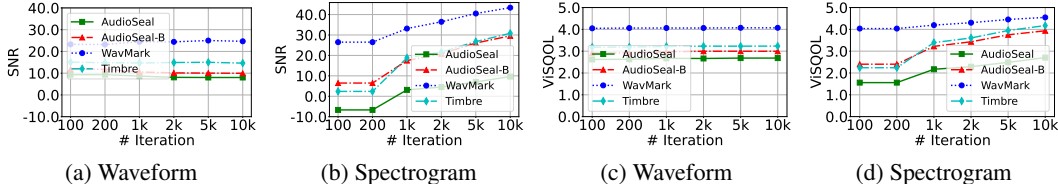

(a) Waveform  (b) Spectrogram  (c) Waveform  (d) Spectrogram

Figure 6: HSJA's audio quality when optimizing watermark-removal perturbations in waveform or spectrogram domain on AudioMarkData. The results on LibriSpeech are in Figure 10 in Appendix.

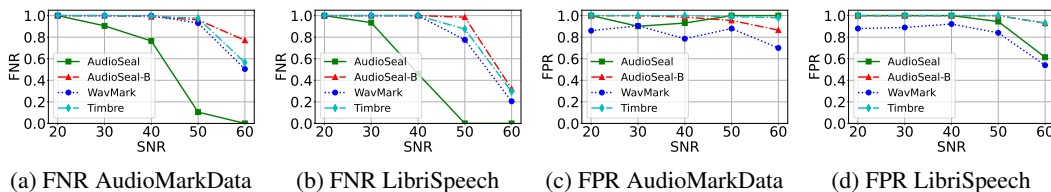

(a) FNR AudioMarkData  (b) FNR LibriSpeech  (c) FPR AudioMarkData  (d) FPR LibriSpeech

Figure 7: Detection results under white-box watermark-removal and watermark-forgery perturbations.

tions. We also observe that such differences may vary across different watermarking methods. For instance, watermarked audios in Esperanto have smaller FNRs on AudioSeal but larger FNRs on WavMark. We hypothesize that Esperanto, as an artificial language, may have specific characteristics (e.g., phonetic patterns, speech dynamics) that interact differently with the watermarking detectors.

Results in Appendix A.8 show detection differences in age groups in terms of FNRs/FPRs. We observe no consistently significant differences across age groups.

## 5.4 Robustness against Black-box Perturbations

We find that audio watermarking methods have good robustness against *existing* black-box watermark-forgery perturbations. In particular, existing black-box watermark-forgery perturbations substantially sacrifice audio quality in order to forge watermarks. However, audio watermarking methods are not robust to *existing* black-box watermark-removal perturbations when an attacker can query the detector API for many times. In particular, they can remove watermarks from watermarked audios while preserving their audio quality given sufficient number of queries to the detector API. When the number of queries to the detector API is limited, audio watermarking methods have good robustness against *existing* black-box watermark-removal perturbations. Next, we discuss results for watermark-removal perturbations found by HSJA, and the results for Square attack are in Appendix A.5.

Recall that HSJA guarantees that the found watermark-removal perturbations are successful while iteratively optimizing them. Therefore, we evaluate the quality of the perturbed watermarked audios when increasing the number of iterations/queries to the watermarking detector. We consider adding perturbations to both waveform and spectrogram domains, and the results are shown in Figure 6. First, in the waveform domain, quality of the perturbed audio does not improve with more iterations, indicating that HSJA struggles to optimize perturbations in the waveform domain. This may be attributed to HSJA's design, which is tailored to attack image classifiers, potentially making it less effective on 1-D audio waveform. Second, in the spectrogram domain, although initial audio quality is inferior to those in the waveform domain, the audio quality improves significantly with more iterations. Specifically, for AudioSeal-B, Timbre, and WavMark, while the SNR/ViSQOL scores are slightly inferior to those in waveform domain under 100 iterations, after 10,000 iterations, the audio quality is considerably better, with WavMark achieving SNR/ViSQOL of 40/4.5, and AudioSeal-B and Timbre reaching approximately 30/4. AudioSeal has better robustness in both waveform and spectrogram domains, maintaining SNR/ViSQOL scores below 10/3. Third, like no-box perturbations, we observe that WavMark is least robust while AudioSeal is the most robust.

We also observe that watermarked audios with attribute "female" are less robust to watermark-removal Square attack perturbations (i.e., have higher FNRs) than those with attribute "male" (see Figure 4b

and more results in Appendix A.7). Like no-box setting, we did not observe robustness gaps among age groups in both black-box and white-box settings (discussed in the next subsection). Moreover, due to computation resource limit, we sampled 200 audio samples in black-box and white-box settings, leading to only 4 samples per language. Therefore, we did not study robustness across languages due to the small-sample issue.

### 5.5 Robustness against White-box Perturbations

Figure 7 shows the detection results under white-box perturbations, where the perturbations are constrained by SNR. We evaluate SNRs from 20 to 60, which correspond to ViSQOL scores from above 3 to 5 (see Figure 8). In other words, our white-box perturbations preserve the audio quality. Our key observation is that existing audio watermarking methods are not robust to white-box watermark-removal and watermark-forgery perturbations. For instance, FNRs reach 1 for all watermarking methods when the SNR of the perturbations is 20 (i.e., ViSQOL of 3.2 and 3.9 on the two datasets). Moreover, all watermarking methods have high FPRs under white-box perturbations that preserve audio quality. We also evaluate iterative Fast Gradient Sign Method (I-FGSM) [11] in Appendix A.11 and get similar conclusion.

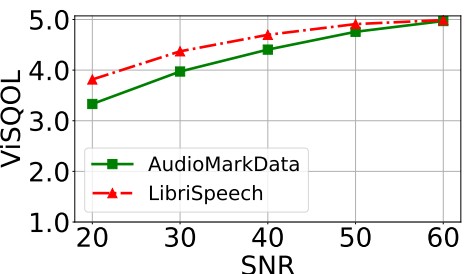

Figure 8: ViSQOL vs. SNR of white-box perturbations.

We also observe that watermarked audios with attribute "female" are less robust to watermark-removal white-box perturbations (i.e., have higher FNRs) than those with attribute "male" (see Figure 22 and more results in Appendix A.7).

## 6 Discussions

**Limitations:** The major limitation of this work is that AudioMarkData contains 25 languages and 4 age groups due to the fact it's sub-sampled from Common-Voice. We deem the collection of audios with more diverse languages and age groups an important future direction.

**Social impacts:** Our AudioMarkBench evaluates the vulnerability of audio watermarks to removal or forgery and has significant implications for the safe usage of audio generation/watermarking techniques. First, watermark removal enables AI-generated audio to be disguised as authentic, potentially fueling misinformation campaigns. Second, watermark forgery allows for false attribution of AI-generated audio, undermining the ability of human creators to protect their work. By assessing the robustness of audio watermarking techniques, our AudioMarkBench contributes to the development of more secure watermarking systems, helping to mitigate the potential negative impacts of AI-generated audio on society.

## 7 Conclusion

In this work, we introduce AudioMarkBench, the first systematic benchmark for evaluating the robustness of audio watermarking against watermark removal/forgery. Our study, involving 3 state-of-the-art methods and 15 perturbation types across 2 datasets (including our new AudioMarkData), reveals that existing watermarking methods lack robustness under various no-box/black-box and white-box perturbations. Additionally, we identify fairness issues, with robustness varying across biological sex and language groups under certain perturbations. Our benchmark promotes further research to enhance robustness and fairness in audio watermarking.

## Acknowledgements

We thank the anonymous reviewers for their constructive comments. This work was supported by NSF under Grant No. 2414406.

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

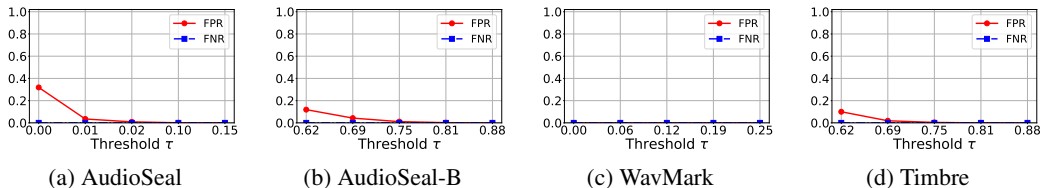

(a) AudioSeal        (b) AudioSeal-B        (c) WavMark        (d) Timbre

Figure 9: Detection results under no perturbations on LibriSpeech. We set the detection threshold $\tau$ for each watermarking method as follows: AudioSeal $\tau = 0.1$, AudioSeal-B $\tau = 0.875$, WavMark $\tau = 0.0$, and Timbre $\tau = 0.8125$, to achieve FPR $< 0.01$ and FNR $< 0.01$.

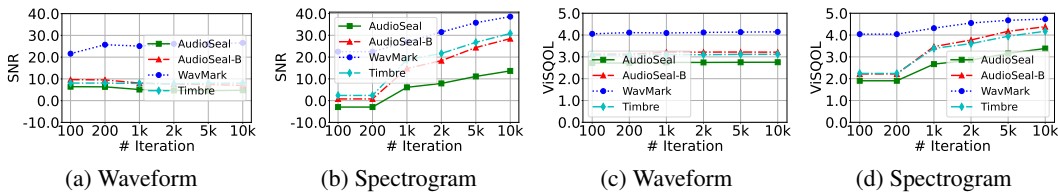

(a) Waveform        (b) Spectrogram        (c) Waveform        (d) Spectrogram

Figure 10: HSJA's audio qualities when optimizing watermark-removal perturbations in waveform or spectrogram domains on LibriSpeech.

## A    Appendix

### A.1    Details of 25 Languages in Our AudioMarkData

EU: Basque, BE: Belarusian, BN: Bengali, YUE: Cantonese, CA: Catalan, ZH-CN: Chinese-China, ZH-HK: Chinese-Hong-Kong, ZH-TW: Chinese-Taiwan, EN: English, EO: Esperanto, FR: French, KA: Georgian, DE: German, HU: Hungarian, IT: Italian, JA: Japanese, LV: Latvian, MHR: Meadow Mari, FA: Persian, RU: Russian, ES: Spanish, SW: Swahili, TA: Tamil, TH: Thai, UK: Ukrainian.

### A.2    Details of No-box Perturbations

We summarize the key parameter, its range, and a brief description for each of the 12 no-box perturbations in Table 2.

Table 2: Details of no-box perturbations.

| Perturbation | Key Parameter $\mathbb{K}$ | Range of $\mathbb{K}$ | Brief Description |
|---|---|---|---|
| Time Stretch | Speed Factor | [0.7, 1.5] | Controls the playback speed of the audio |
| Gaussian Noise | SNR (dB) | [5, 40] | Adds random noise constrained by SNR |
| Background Noise | SNR (dB) | [5, 40] | Adds background noise constrained by SNR |
| SoundStream | # Quantizers | [4, 16] | Neural network-based audio codec |
| Opus | Bitrate (kbps) | [16, 256] | Widely used audio codec |
| EnCodec | Bandwidth (kHz) | [1.5, 24.0] | Neural network-based audio codec |
| Quantization | Bit levels | [4, 64] | Converts audio signal to $n$ bit level discrete values |
| Highpass Filter | Cutoff Ratio | [0.1, 0.5] | Filters out low frequency banks |
| Lowpass Filter | Cutoff Ratio | [0.1, 0.5] | Filters out high frequency banks |
| Smooth | Window Size | [6, 22] | Applies a Gaussian smooth effect using 1-D convolution |
| Echo | Delay (sec) | [0.1, 0.9] | Adds a decayed and delayed replay |
| MP3 Compression | Bitrate (kbps) | [8, 40] | Widely used audio codec |

### A.3    Details of Black-box Perturbations

**HSJA:**  Given an audio waveform $s$, to perform the Hop Skip Jump Attack (HSJA), an initial adversarial example $s + \delta$ must be provided, where $\delta$ is sampled using Gaussian noise in our

experiment. We employ a greedy algorithm to determine the Gaussian noise that achieves the maximum SNR while still successfully evading detection, ensuring that $\delta$ is of minimal size.

For attacks directly targeting the audio waveform, $s + \delta$ is used as the input, and $\delta$ is optimized using the HSJA strategy. For attacks on the spectrogram, $s + \delta$ is first transformed into a spectrogram $C_{s+\delta} = (a_{s+\delta}, p_{s+\delta})$, where $a$ and $p$ represent the amplitude and phase, respectively. The attacker then uses this spectrogram as the input for the attack.

Regarding gradient estimation within the HSJA algorithm, we initialize the number of estimations at 100 and set a maximum limit of 1,000 estimations. The attack proceeds through 10,000 iterations, maintaining other parameters at their default settings as specified by the HSJA method.

**Square attack:** The Square attack is specifically designed to perform adversarial attacks on image classifiers. Given a perturbation size, it leverages this boundary to try to evade the detector by lowering the detection confidence. (In our setting, it is the global detection probability $P_S$ or the bitwise accuracy between the decoded and ground truth watermark.) The attack designs two individual algorithms for optimizing based on $\ell_2$ and $\ell_\infty$ perturbations. We conducted experiments on both algorithms but only found the attack based on $\ell_\infty$ is effective. For optimizing $\ell_\infty$ perturbations, the attack first crafts several vertical stripes as perturbations, then adds square-shaped perturbations to perform the random search. Given its nature of attacking images, we extend it to attack the spectrogram. To maintain uniformity, we also run 10,000 iterations and keep the parameters the same as the default settings.

### A.4 Sovling Optimization Problems in White-box Perturbations

Given a watermarked/unwatermarked audio waveform $s_w/s_u$, white-box performs watermark removal/forgery by optimizing a perturbation $\delta$ added to the audio waveform. Specifically, let $s \in \mathbb{R}^T$ be the waveform in length $T$, in white-box setting, we optimize the perturbation $\delta \in \mathbb{R}^T$ to achieve watermark removal/forgery. Detailed algorithms are shown in Algorithm 3 and Algorithm 4.

---

**Algorithm 1** White-box loss: $\ell(s, w)$

---

**Input:** audio $s \in \mathbb{R}^T$, ground truth watermark $w \in \{0, 1\}^n$, decoder $\texttt{Dec}$, detection threshold $\tau$
**Output:** Loss $\ell(s, w)$
1: **if** use *AudioSeal* **then**
2:      $P_s \leftarrow \texttt{Dec}(s)$             $\triangleright$ global detection probability
3:      **return** $\ell = \max(0, P_s - \tau)$
4: **else**             $\triangleright$ use AudioSeal-B, Timbre, or Wavmark
5:      decoded watermark $\texttt{Dec}(s)$
6:      **return** $\ell = -\sum_{i=1}^{n} w_i \log(\texttt{Dec}(s)_i) + (1 - w_i) \log(1 - \texttt{Dec}(s)_i)$
7: **end if**

---

**Algorithm 2** Compute Scaling Factor $f_R(s, \delta, R)$

---

**Input:** Signal $s \in \mathbb{R}^T$, perturbation $\delta \in \mathbb{R}^T$, preset SNR $R$
**Output:** Scaling factor $r$
1: $P_s \leftarrow \sum_{i=1}^{T} s_i^2 / T$             $\triangleright$ signal power
2: $P_\delta \leftarrow \sum_{i=1}^{T} \delta_i^2 / T$             $\triangleright$ noise power
3: $snr \leftarrow 10 \cdot \log_{10}(P_s / P_\delta)$
4: **if** $snr < R$ **then**             $\triangleright$ Need rescaling
5:      $r \leftarrow 10^{(R-snr)/10}$
6: **else**
7:      $r \leftarrow 1$
8: **end if**
9: **return** $r$

---

---

**Algorithm 3** Optimizing White-box Watermark-removal Perturbations

---

**Input:** Watermarked audio $s_w \in \mathbb{R}^T$, ground truth watermark $w \in \{0,1\}^n$, watermarking decoder `Dec`, detection threshold $\tau$, SNR restriction $R$, iteration *iter*, learning rate $\alpha$

**Output:** Optimal perturbation $\hat{\delta}$

1: $\delta \leftarrow \mathbf{0} \in \mathbb{R}^T$                                                 ▷ Initialize perturbation
2: $\hat{\delta} \leftarrow \delta$
3: **if** use *AudioSeal* **then**                                                 ▷ Initial optimization function
4:     $\mathcal{Q}(\cdot) \leftarrow P_s(\cdot)$
5: **else**                                                 ▷ use AudioSeal-B, Timbre, or Wavmark
6:     $\mathcal{Q}(\cdot) \leftarrow \texttt{BA}(\texttt{Dec}(\cdot), w)$
7: **end if**
8: $\hat{\mathcal{Q}} \leftarrow \mathcal{Q}(s_w)$
9: **for** $i \leftarrow 1$ to *iter* **do**
10:     $\delta \leftarrow \delta - \alpha \cdot \nabla_\delta \ell(s_w, \neg w)$                                                 ▷ loss returned by Algorithm 1
11:     $r \leftarrow f_R(s_w, \delta, R)$                                                 ▷ scaling factor returned by Algorithm 2
12:     **if** $r > 1$ **then**
13:         $\delta \leftarrow \delta / r$
14:     **end if**
15:     **if** $\hat{\mathcal{Q}} > \mathcal{Q}(s_w + \delta)$ **then**
16:         $\hat{\delta} \leftarrow \delta$
17:         $\hat{\mathcal{Q}} \leftarrow \mathcal{Q}(s_w + \delta)$
18:     **end if**
19:     **if** $\hat{\mathcal{Q}} \leq \tau$ **then**                                                 ▷ early stopping
20:         **return** $\hat{\delta}$
21:     **end if**
22: **end for**
23: **return FAIL**

---

---

**Algorithm 4** Optimizing White-box Watermark-forgery Perturbations

---

**Input:** Unwatermarked audio $s_u \in \mathbb{R}^T$, forgery watermark $w_f \in \{0,1\}^n$, decoder `Dec`, detection threshold $\tau$, SNR restriction $R$, iteration *iter*, learning rate $\alpha$

**Output:** Optimal perturbation $\hat{\delta}$

1: $\delta \leftarrow \mathbf{0} \in \mathbb{R}^T$                                                 ▷ Initialize perturbation
2: $\hat{\delta} \leftarrow \delta$
3: **if** use *AudioSeal* **then**                                                 ▷ Initial optimization function
4:     $\mathcal{Q}(\cdot) \leftarrow P_s(\cdot)$
5: **else**                                                 ▷ use AudioSeal-B, Timbre, or Wavmark
6:     $\mathcal{Q}(\cdot) \leftarrow \texttt{BA}(\texttt{Dec}(\cdot), w_f)$
7: **end if**
8: $\hat{\mathcal{Q}} \leftarrow \mathcal{Q}(s_u)$
9: **for** $i \leftarrow 1$ to *iter* **do**
10:     $\delta \leftarrow \delta - \alpha \cdot \nabla_\delta \ell(s_u, w_f)$                                                 ▷ loss returned by Algorithm 1
11:     $r \leftarrow f_R(s_u, \delta, R)$                                                 ▷ scaling factor returned by Algorithm 2
12:     **if** $r > 1$ **then**
13:         $\delta \leftarrow \delta / r$
14:     **end if**
15:     **if** $\hat{\mathcal{Q}} < \mathcal{Q}(s_u + \delta)$ **then**
16:         $\hat{\delta} \leftarrow \delta$
17:         $\hat{\mathcal{Q}} \leftarrow \mathcal{Q}(s_u + \delta)$
18:     **end if**
19:     **if** $\hat{\mathcal{Q}} > \tau$ **then**                                                 ▷ early stopping
20:         **return** $\hat{\delta}$
21:     **end if**
22: **end for**
23: **return FAIL**

---

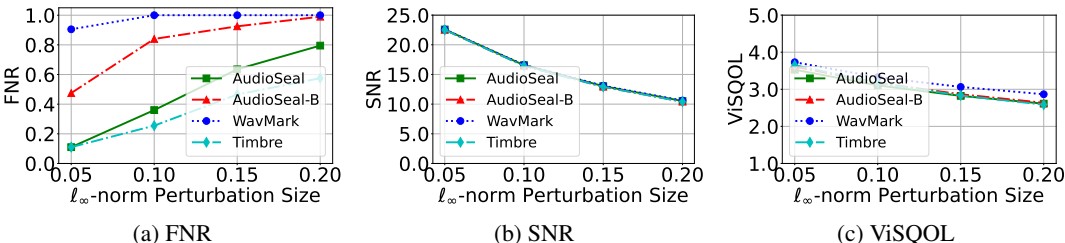

(a) FNR  (b) SNR  (c) ViSQOL

Figure 11: Square attack results of watermark-removal perturbations on AudioMarkData.

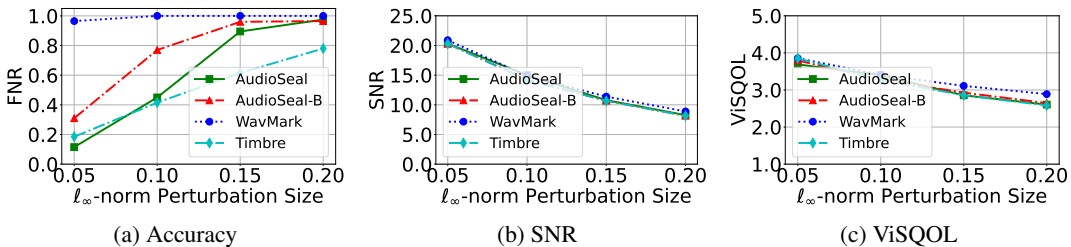

(a) Accuracy  (b) SNR  (c) ViSQOL

Figure 12: Square attack results of watermark-removal perturbations on LibriSpeech.

## A.5 Results for Square Attack

Figure 11 and Figure 12 shows the FNR results of three watermarking methods under Square attack perturbations on our AudioMarkData and LibriSpeech, respectively. We observe that

## A.6 Results for No-box Perturbations

Figure 13, Figure 15, Figure 14, Figure 14 show FPR/FNR, SNR, and ViSQOL results on AudioMark-Data and LibriSpeech under 11 no-box perturbations. We observe that SoundStream and Opus are also effective watermark-removal no-box perturbations that can preserve good quality for original watermarked audios via achieving high FNRs as well as ViSQOL scores higher than 3. Quantization is an effective watermark-forgery no-box perturbation that achieves high FPRs while preserving ViSQOL scores closed to 3.

## A.7 Detection Differences across Biological Sexes

Figure 17, Figure 18, Figure 19, Figure 20, Figure 21 and Figure 22 show the FNRs across biological sex groups among different models and various perturbations. We observe significant differences of robustness gaps between "female" and "male" biological sex groups. In our experiments, we find that there is no evidence for significant differences across biological sexes under HSJA perturbations.

## A.8 Detection Differences across Age

Figure 23, Figure 24, Figure 25 show the FNRs on some effective no-box watermark-removal perturbations. Figure 26 and Figure 27 show the FPRs on some effective no-box watermark-forgery perturbations. We do not observe significant differences of robustness gaps among age groups persist across all watermarking methods. For those settings having statistically significant differences in terms of robustness gaps for age groups, we report their $p$-values in Table 3 and Table 4.

Table 3: Two-tail test results for FNRs in different age groups against watermark-removal perturbations. We consider significance level $\alpha = 0.05$.

| | AudioSeal | AudioSeal-B | Timbre | WavMark |
|---|---|---|---|---|
| Gaussian noice | 8.52e-12 (twenties, forties) | 1.74e-3 (thirties, fourties) | 2.08e-3 (twenties, forties) | / |
| EnCodeC | 1.36e-6 (twenties, forties) | / | / | / |
| Opus | / | / | / | / |

Table 4: Two-tail test results for FPRs in different age groups against watermark-forgery perturbations. We consider significance level $\alpha = 0.05$.

| | AudioSeal | AudioSeal-B | Timbre | WavMark |
|---|---|---|---|---|
| EnCodeC | 1.74e-4 (twenties, fourties) | / | / | / |
| Quantization | / | 3.83e-2 (teens, thirties) | / | / |

## A.9 Languages Differences against Watermark-removal Perturbations

Figure 28, Figure 29, Figure 30 and Figure 31 show results of languages differences against watermark-removal perturbations when using three audio watermarking methods. We observe significant difference on robustness gaps against some watermark-removal differences among different languages.

## A.10 Results on FMA Music Dataset

We conducted additional experiments using the FMA music dataset [6]. Table 5 shows the results without any perturbation; Table 6 shows the results under several no-box perturbations; Table 7 shows the results under black-box attacks; and Table 8 and Table 9 show the results under white-box removal and forgery attacks.

Table 5: Detection results under no perturbations on FMA.

(a) AudioSeal

| Tau | 0.01 | 0.02 | 0.05 | 0.1 | 0.15 |
|---|---|---|---|---|---|
| FPR | 0.210 | 0.130 | 0.060 | 0.030 | 0.020 |
| FNR | 0.000 | 0.000 | 0.000 | 0.000 | 0.000 |

(b) AudioSeal-B

| Tau | 0.625 | 0.6875 | 0.75 | 0.8125 | 0.875 |
|---|---|---|---|---|---|
| FPR | 0.080 | 0.010 | 0.010 | 0.000 | 0.000 |
| FNR | 0.000 | 0.000 | 0.010 | 0.040 | 0.110 |

(c) Timbre

| Tau | 0.625 | 0.6875 | 0.75 | 0.8125 | 0.875 |
|---|---|---|---|---|---|
| FPR | 0.100 | 0.000 | 0.000 | 0.000 | 0.000 |
| FNR | 0.000 | 0.000 | 0.000 | 0.010 | 0.030 |

(d) WavMark

| Tau | 0.0 | 0.0625 | 0.125 | 0.1875 | 0.25 |
|---|---|---|---|---|---|
| FPR | 0.000 | 0.000 | 0.000 | 0.000 | 0.000 |
| FNR | 0.000 | 0.000 | 0.000 | 0.000 | 0.000 |

Table 6: Detection results (FNR/FPR) under background noise and time stretch on FMA.

(a) Background noise

| SNR | AudioSeal | AudioSeal-B | Timbre | WavMark |
|---|---|---|---|---|
| 5 | .03/.01 | .44/.00 | .52/.00 | .96/.00 |
| 10 | .00/.02 | .26/.00 | .25/.00 | .68/.00 |
| 20 | .00/.01 | .04/.00 | .04/.00 | .07/.00 |
| 30 | .00/.01 | .02/.01 | .02/.00 | .00/.00 |
| 40 | .00/.01 | .01/.00 | .01/.00 | .00/.00 |

(b) Time stretch

| Stretch | AudioSeal | AudioSeal-B | Timbre | WavMark |
|---|---|---|---|---|
| 0.7 | .60/.01 | .24/.01 | .08/.00 | .41/.00 |
| 0.9 | .45/.02 | .19/.00 | .04/.00 | .20/.00 |
| 1.1 | .25/.02 | .10/.01 | .04/.00 | .24/.00 |
| 1.3 | .57/.01 | .17/.01 | .12/.00 | .57/.00 |
| 1.5 | .73/.02 | .31/.02 | .13/.00 | .84/.00 |

Table 7: Results under black-box Square attack on FMA.

(a) AudioSeal

| $\ell_\infty$ Bound | FNR | SNR | ViSQOL |
|---|---|---|---|
| 0.05 | 0.0 | 24.85 | 4.57 |
| 0.1 | 0.0 | 18.84 | 4.33 |
| 0.15 | 0.0 | 15.33 | 4.14 |
| 0.2 | 0.0 | 12.83 | 3.96 |

(b) AudioSeal-B

| $\ell_\infty$ Bound | FNR | SNR | ViSQOL |
|---|---|---|---|
| 0.05 | 0.15 | 24.85 | 4.55 |
| 0.1 | 0.25 | 18.83 | 4.29 |
| 0.15 | 0.45 | 15.33 | 4.08 |
| 0.2 | 0.55 | 12.85 | 3.91 |

(c) Timbre

| $\ell_\infty$ Bound | FNR | SNR | ViSQOL |
|---|---|---|---|
| 0.05 | 0.0 | 25.68 | 4.79 |
| 0.1 | 0.07 | 19.66 | 4.58 |
| 0.15 | 0.14 | 16.14 | 4.39 |
| 0.2 | 0.21 | 13.66 | 4.22 |

(d) WavMark

| $\ell_\infty$ Bound | FNR | SNR | ViSQOL |
|---|---|---|---|
| 0.05 | 0.0 | 25.96 | 4.81 |
| 0.1 | 0.67 | 19.63 | 4.70 |
| 0.15 | 1.0 | 16.18 | 4.58 |
| 0.2 | 1.0 | 14.52 | 4.52 |

Table 8: Detection results under white-box removal attack on FMA.

(a) AudioSeal

| SNR | 20 | 30 | 40 | 50 | 60 |
|---|---|---|---|---|---|
| FNR | 1.00 | 0.85 | 0.35 | 0.00 | 0.00 |

(b) AudioSeal-B

| SNR | 20 | 30 | 40 | 50 | 60 |
|---|---|---|---|---|---|
| FNR | 1.00 | 1.00 | 0.95 | 0.50 | 0.00 |

(c) Timbre

| SNR | 20 | 30 | 40 | 50 | 60 |
|---|---|---|---|---|---|
| FNR | 1.00 | 0.95 | 0.85 | 0.45 | 0.10 |

(d) WavMark

| SNR | 20 | 30 | 40 | 50 | 60 |
|---|---|---|---|---|---|
| FNR | 1.00 | 1.00 | 1.00 | 0.50 | 0.40 |

Table 9: Detection results under white-box forgery attack on FMA.

(a) AudioSeal

| SNR | 20 | 30 | 40 | 50 | 60 |
|---|---|---|---|---|---|
| FPR | 1.00 | 1.00 | 1.00 | 1.00 | 1.00 |

(b) AudioSeal-B

| SNR | 20 | 30 | 40 | 50 | 60 |
|---|---|---|---|---|---|
| FPR | 1.00 | 0.95 | 0.90 | 0.40 | 0.30 |

(c) Timbre

| SNR | 20 | 30 | 40 | 50 | 60 |
|---|---|---|---|---|---|
| FPR | 1.00 | 1.00 | 0.90 | 0.50 | 0.20 |

(d) WavMark

| SNR | 20 | 30 | 40 | 50 | 60 |
|---|---|---|---|---|---|
| FPR | 1.00 | 1.00 | 1.00 | 1.00 | 1.00 |

## A.11 More Results on AudioMarkData

We applied I-FGSM as an additional white-box attack in Table 10. In Table 11, we show the results for composed no-box perturbations including EnCodeC with 24kHz, MP3 with 16kbps, and Gaussian noise with SNR of 20dB.

Table 10: Results for I-FGSM on AudioMarkData.

(a) Watermark removal (FNR)

| SNR | AudioSeal | AudioSeal-B | Timbre | WavMark |
|---|---|---|---|---|
| 20 | 1.00 | 1.00 | 1.00 | 1.00 |
| 30 | 0.75 | 1.00 | 1.00 | 1.00 |
| 40 | 0.25 | 0.90 | 0.85 | 0.50 |
| 50 | 0.00 | 0.40 | 0.45 | 0.50 |
| 60 | 0.00 | 0.15 | 0.05 | 0.15 |

(b) Watermark forgery (FPR)

| SNR | AudioSeal | AudioSeal-B | Timbre | WavMark |
|---|---|---|---|---|
| 20 | 1.00 | 1.00 | 1.00 | 1.00 |
| 30 | 1.00 | 1.00 | 1.00 | 1.00 |
| 40 | 1.00 | 0.75 | 1.00 | 1.00 |
| 50 | 1.00 | 0.20 | 0.90 | 1.00 |
| 60 | 1.00 | 0.00 | 0.50 | 1.00 |

Table 11: Results for composed no-box perturbations.

(a) EnCodeC + MP3

| Method | FNR | FPR |
|---|---|---|
| AudioSeal | 0.99 | 0.00 |
| AudioSeal-B | 1.00 | 0.00 |
| Timbre | 0.99 | 0.00 |
| WavMark | 1.00 | 0.00 |

(b) MP3 + EnCodeC

| Method | FNR | FPR |
|---|---|---|
| AudioSeal | 0.95 | 0.00 |
| AudioSeal-B | 1.00 | 0.00 |
| Timbre | 0.96 | 0.00 |
| WavMark | 1.00 | 0.00 |

(c) Gaussian noise + MP3

| Method | FNR | FPR |
|---|---|---|
| AudioSeal | 0.09 | 0.00 |
| AudioSeal-B | 0.65 | 0.00 |
| Timbre | 0.38 | 0.00 |
| WavMark | 1.00 | 0.00 |

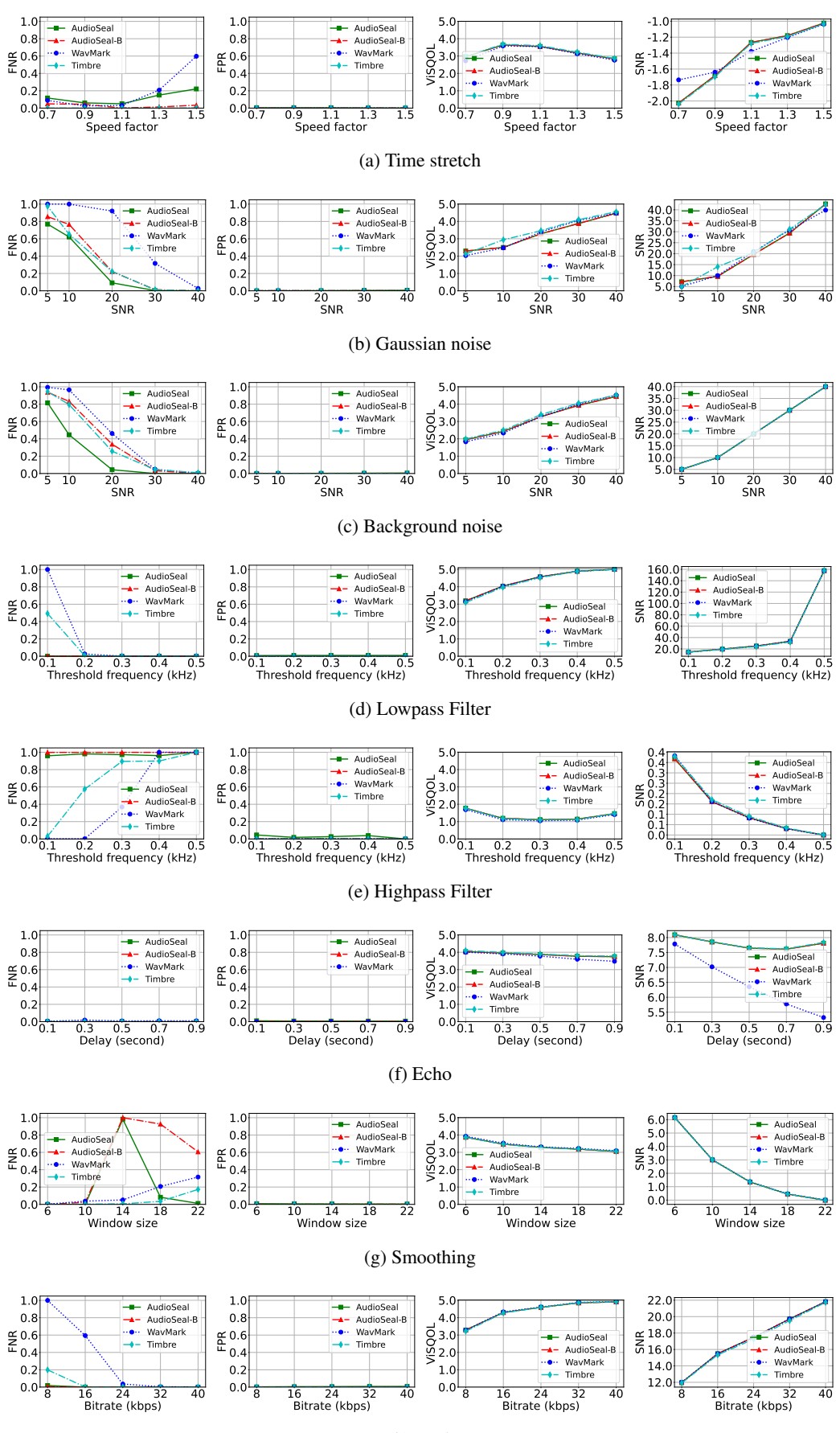

Figure 13: Detection results under eight watermark-removal no-box perturbations on AudioMarkData.

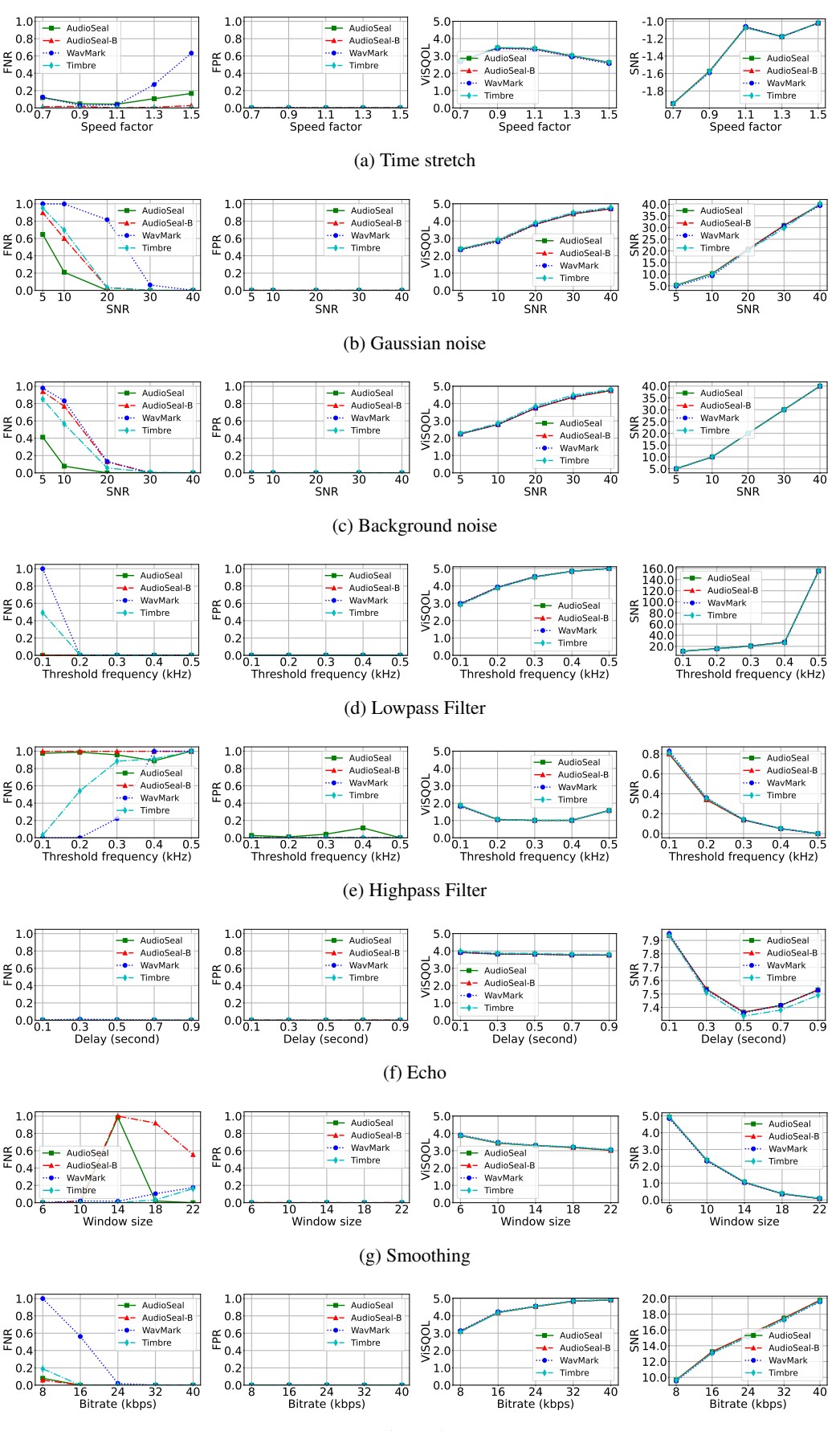

Figure 14: Detection results under eight watermark-removal no-box perturbations on LibriSpeech.

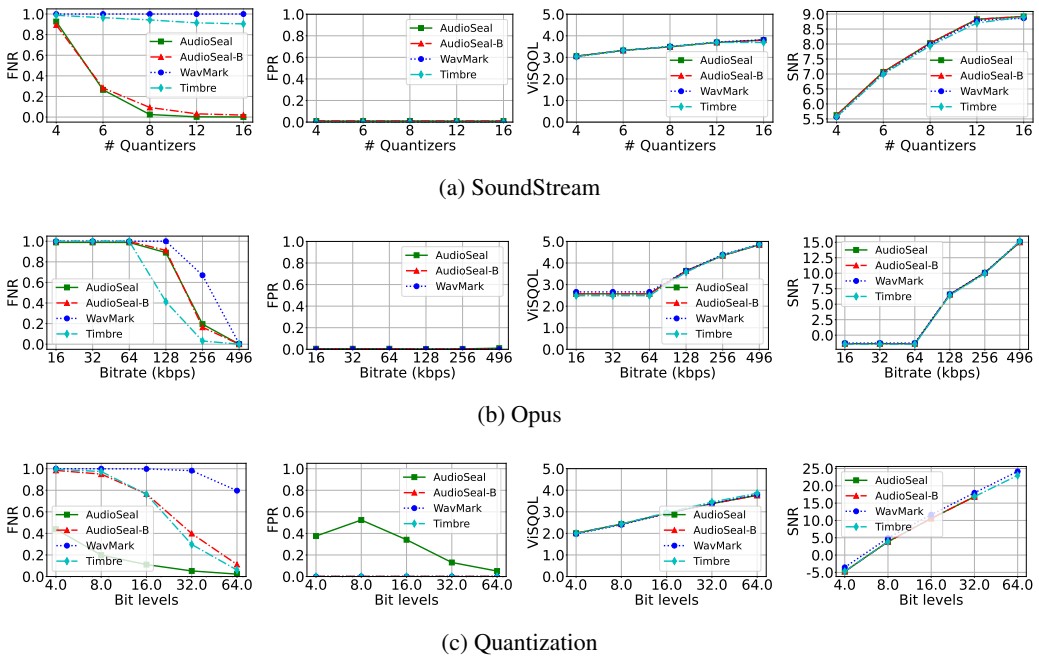

Figure 15: Detection results under another three watermark-removal no-box perturbations on AudioMarkData.

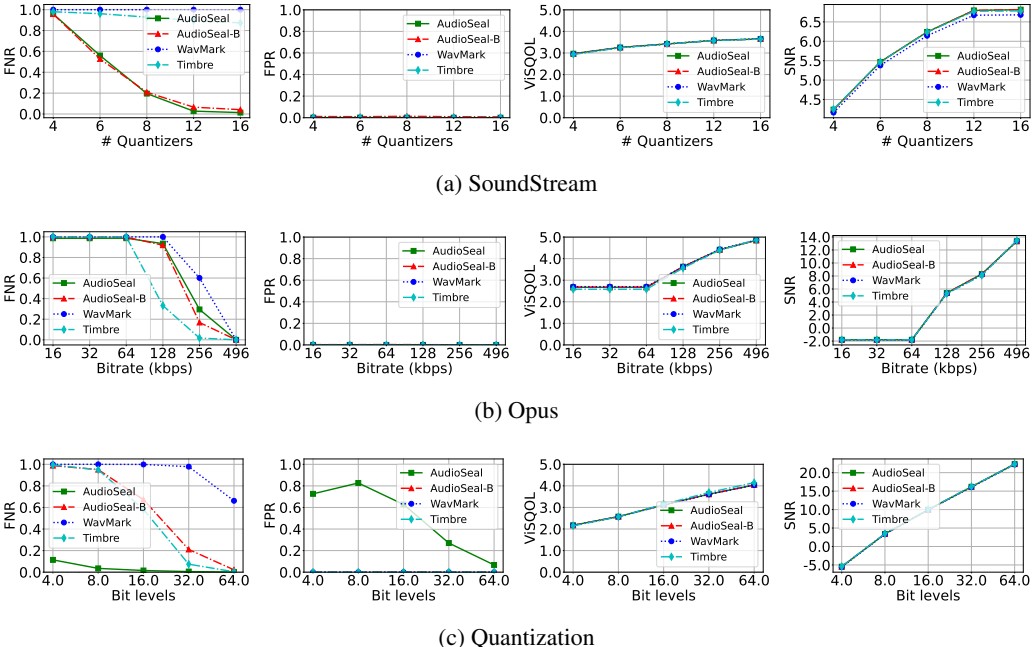

Figure 16: Detection results under another three watermark-removal no-box perturbations on LibriSpeech.

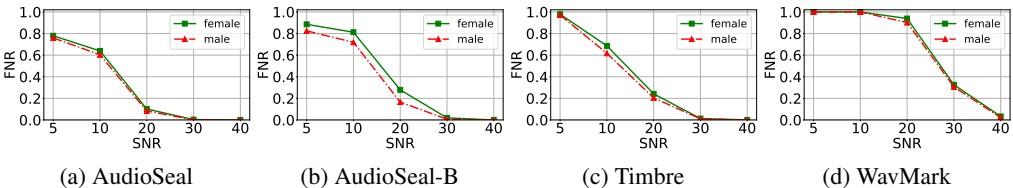

Figure 17: FNRs in biological sexes against watermark-removal Gaussian noise perturbations.

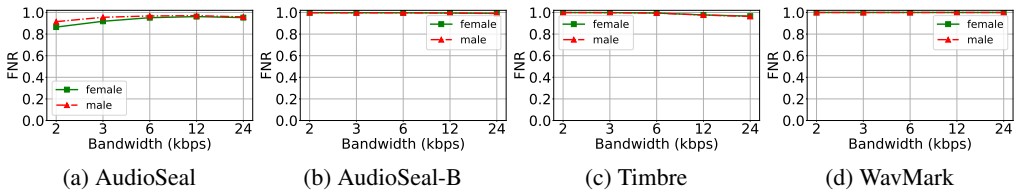

Figure 18: FNRs in biological sexes against watermark-removal EnCodeC perturbations.

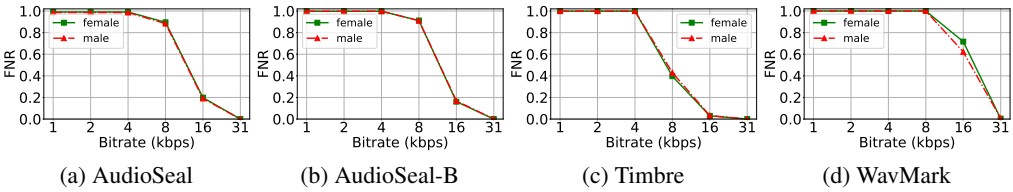

Figure 19: FNRs in biological sexes against watermark-removal Opus perturbations.

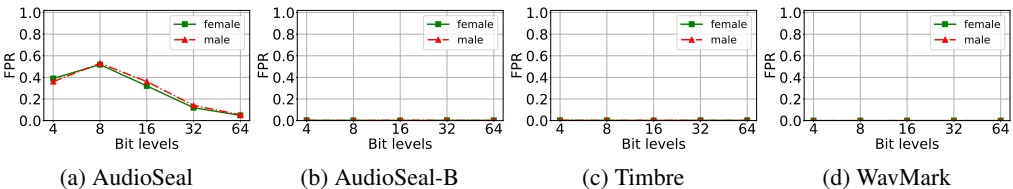

Figure 20: FPRs in biological sexes against watermark-removal Quantization perturbations.

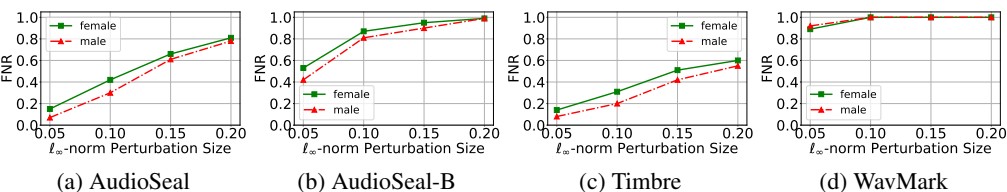

Figure 21: FNRs in biological sexes against watermark-removal Square attack perturbations.

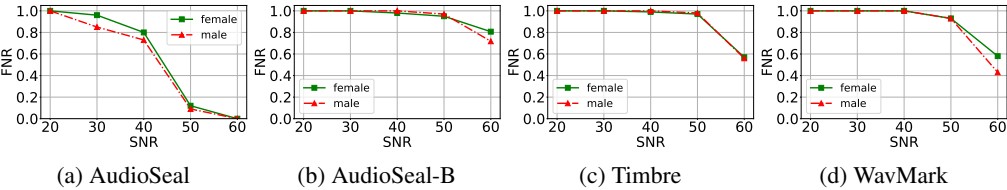

Figure 22: FNRs in biological sexes against watermark-removal white-box perturbations.

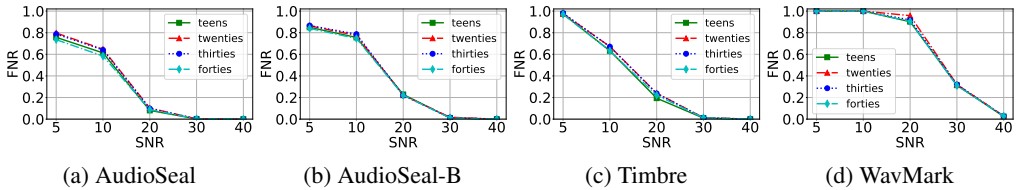

(a) AudioSeal     (b) AudioSeal-B     (c) Timbre     (d) WavMark

Figure 23: FNRs in different age groups against watermark-removal Gaussian noise perturbations.

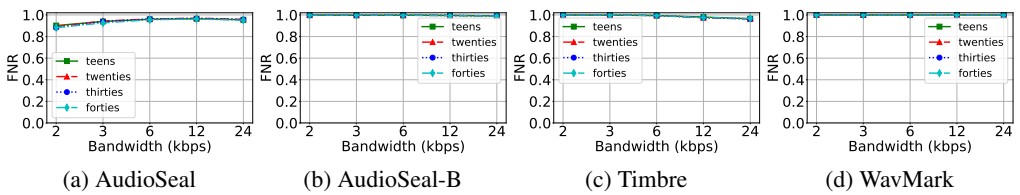

(a) AudioSeal     (b) AudioSeal-B     (c) Timbre     (d) WavMark

Figure 24: FNRs in different age groups against watermark-removal EnCodeC perturbations.

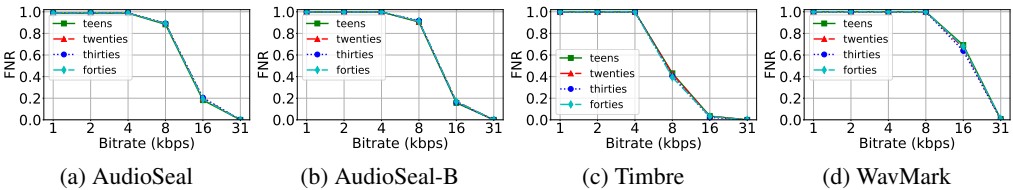

(a) AudioSeal     (b) AudioSeal-B     (c) Timbre     (d) WavMark

Figure 25: FNRs in different age groups against watermark-removal Opus perturbations.

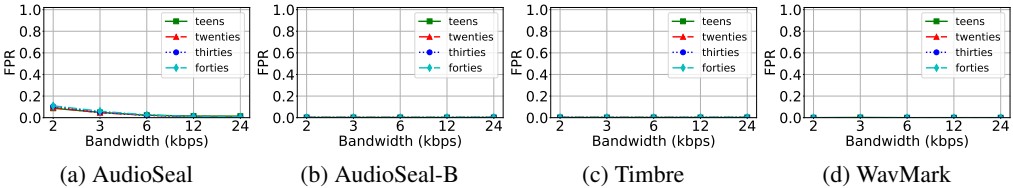

(a) AudioSeal     (b) AudioSeal-B     (c) Timbre     (d) WavMark

Figure 26: FPRs in different age groups against watermark-forgery EnCodeC perturbations.

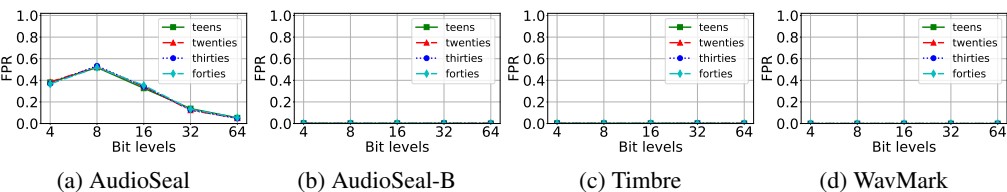

(a) AudioSeal     (b) AudioSeal-B     (c) Timbre     (d) WavMark

Figure 27: FPRs in different age groups against watermark-forgery Quantization perturbations.

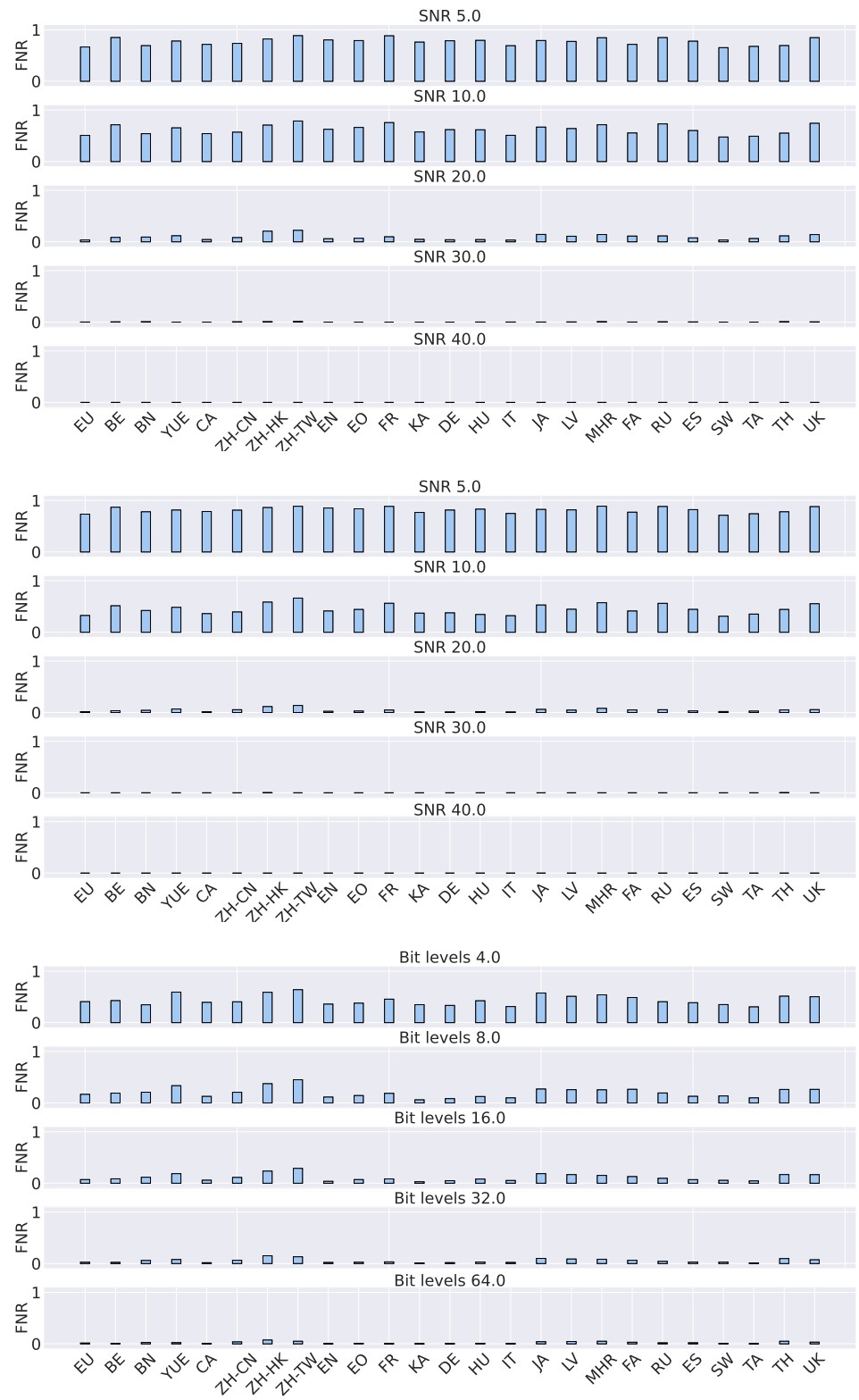

Figure 28: Language difference against watermark-removal perturbations. The watermarking method is AudioSeal. **Upper:** Gaussian noise, **Middle:** Background noise, **Lower:** Quantization.

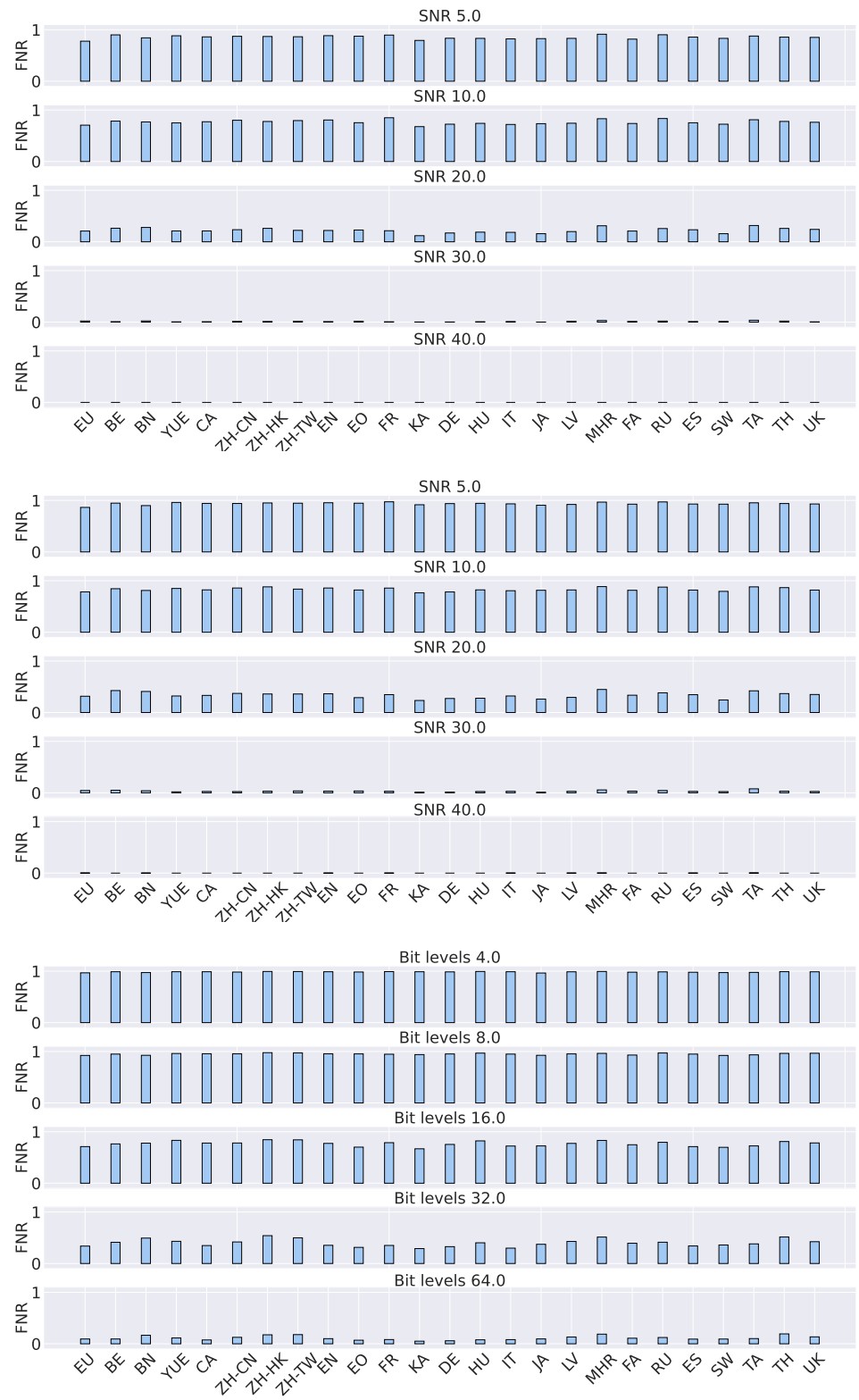

Figure 29: Language difference against watermark-removal perturbations. The watermarking method is AudioSeal-B. **Upper:** Gaussian noise, **Middle:** Background noise, **Lower:** Quantization.

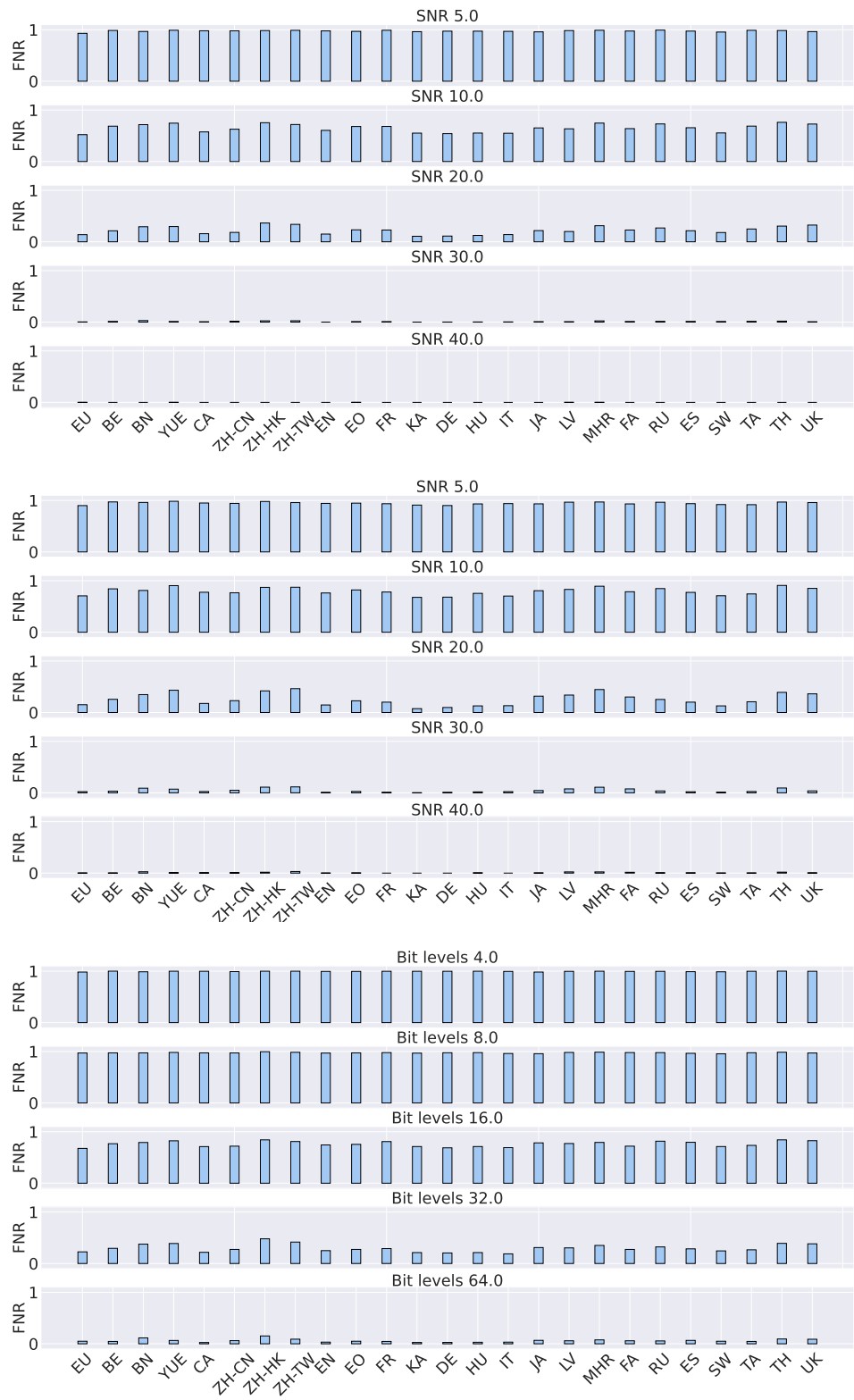

Figure 30: Language difference against watermark-removal perturbations. The watermarking method is Timbre. **Upper:** Gaussian noise, **Middle:** Background noise, **Lower:** Quantization.

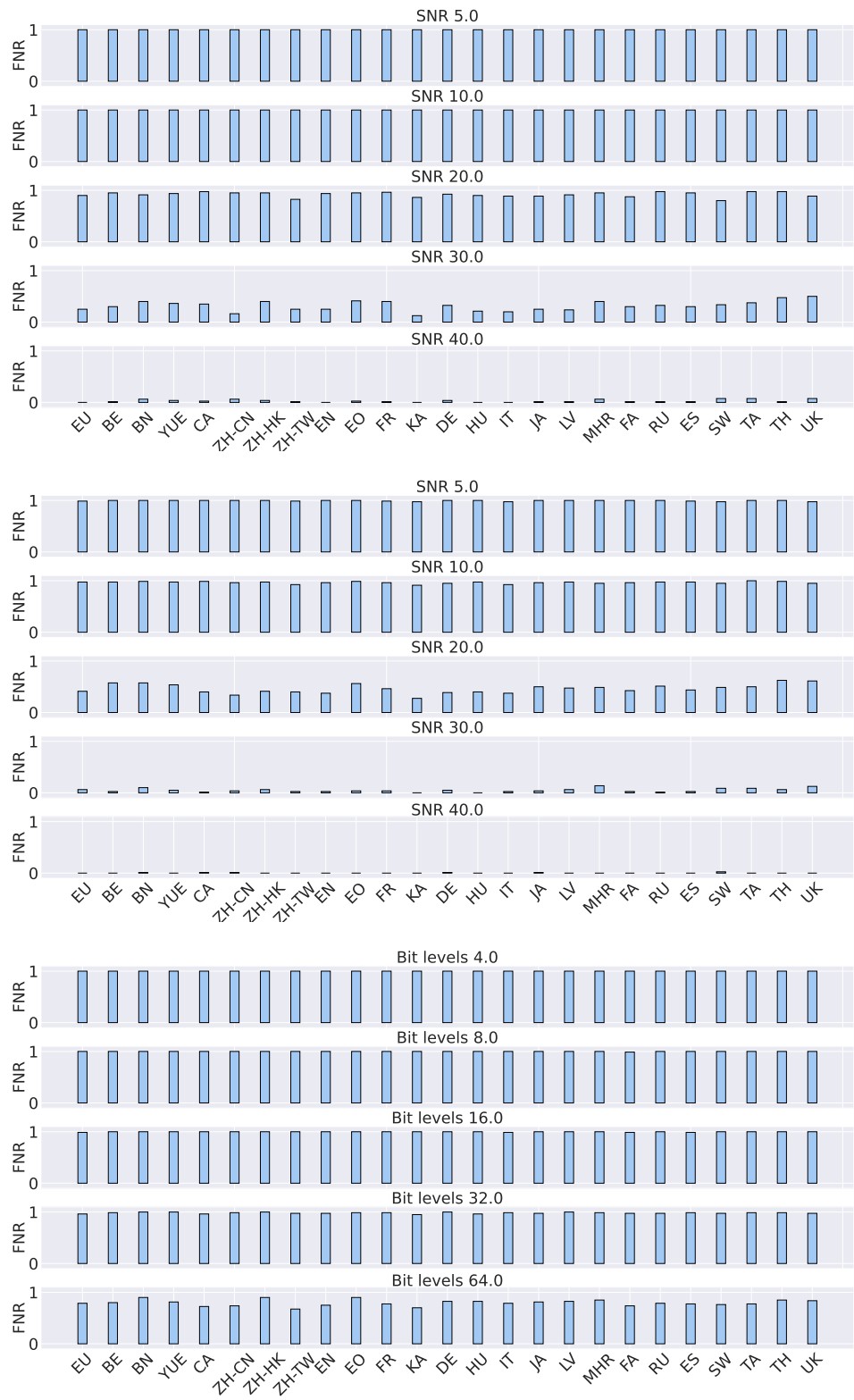

Figure 31: Language difference against watermark-removal perturbations. The watermarking method is WavMark. **Upper:** Gaussian noise, **Middle:** Background noise, **Lower:** Quantization.

