# OpenReview forum: "AudioMarkBench: Benchmarking Robustness of Audio Watermarking"
_NeurIPS.cc/2024/Datasets_and_Benchmarks_Track — NeurIPS 2024 Track Datasets and Benchmarks Poster_

### Official Review · Reviewer_M2hh · 2024-07-09
**Probably the first audio watermarking benchmark**

**Rating:** 7
**Confidence:** 4
**Clarity:** The paper is well written and organiz…

**Review:**

1. The paper only includes 3 audio watermarking methods. More methods
should be considered to provide a more comprehensive evaluation of
current audio watermarking methods.
2. As pointed out in the paper, the constructed AudioMarkData still lacks of
diversity in terms of language and age groups.
3. The observed disparity in robustness between genders is notable, as it
appears in most evaluation scenarios compared to other attributes. What
might cause this fairness gap? Is it due to the training data for audio watermarking methods, or some inherent differences in male and female
voice? This issue is non-trivial and often overlooked? The paper would be
further improved by providing additional insights into these potential causes
and solutions.

**Strengths:**

1. This paper conducts extensive experiments and provides a comprehensive
evaluation of the robustness of audio watermarking methods against
various kinds of watermark-removal and watermark-forgery perturbations.
2. The introduction of AudioMarkData, which includes a diverse of audio
samples across languages, genders and ages, provides a valuable
resources for future research and development in audio watermarking.
3. The evaluation on the proposed dataset uncovers the vulnerabilities of
current watermarking techniques and identifies the fairness issues across
languages, genders and age groups, highlighting the need for more robust
and fairness-aware audio watermarking methods

**Additional Feedback:**

N/A

**Correctness:**

The proposed dataset is constructed in a sound way. The evaluation is
comprehensive.

**Documentation:**

The paper provides sufficient details on dataset and codes for reproducibility.

**Ethics:**

No, there are no or only very minor ethics concerns

**Limitations:**

The authors adequately addressed the limitations and potential negative societal impact of their work.

**Opportunities For Improvement:**

See Review section.

**Relation To Prior Work:**

This paper seems the first work of benchmarking the robustness of audio
watermarking methods.

**Summary And Contributions:**

This paper presents AudioMarkBench, a benchmark framework that evaluates
the robustness of 3 state-of-the-art audio watermarking methods against 15
types of perturbations in no-box, white-box, and black-box settings.
Additionally, it introduces a new dataset that spans various languages,
genders and age groups for evaluation. The results reveal the current
limitation in audio watermarking techniques and offer several insightful
findings.

---

> ### Author Rebuttal · Authors · 2024-08-19
>
> We're grateful for the reviewer's helpful feedback and positive score. Please see our responses to their questions and suggestions below.
>
> ### Comment-1: More audio watermarking methods.
> ### Response-1:
> We appreciate your feedback! Our current focus is on three state-of-the-art audio watermarking methods, and we've designed our benchmark and evaluation code to be flexible and adaptable for future additions. If you have particular watermarking methods you'd like us to explore, please don't hesitate to suggest them, and we will be happy to include them in the revision.
>
>
> ### Comment-2: Lacks of diversity in terms of language and age groups.
> ### Response-2:
> Common-Voice is the only audio dataset available that includes labels for both language and age groups. We've made our best effort to collect and categorize audio examples from Common-Voice, covering 25 languages, biological sexes, and age groups. Manually labeling additional audio examples for diverse languages is time-consuming, and distinguishing age groups without provided labels is particularly challenging.
>
> ### Comment-3: Disparity in robustness between genders.
> ### Response-3:
> Thanks for pointing this out! We agree that the observed disparity in robustness between genders is likely influenced by the training data used for audio watermarking methods. For instance, AudioSeal is trained on the VoxPopuli [1] dataset, where, as indicated in Table 1 of the original VoxPopuli paper, less than 50% of the training data represents female voices. We will include this analysis in our paper to address this issue.
>
> ### Reference
> [1] Wang, Changhan, et al. "VoxPopuli: A large-scale multilingual speech corpus for representation learning, semi-supervised learning and interpretation." NeurIPS, 2021.

---

> > ### Comment · Reviewer_M2hh · 2024-08-23
> >
> > Thanks for the response. I am keeping the current rating.

---

### Official Review · Reviewer_1ur8 · 2024-07-16
**Review for AudioMarkBench: Benchmarking Robustness of Audio Watermarking**

**Rating:** 7
**Confidence:** 4
**Correctness:** The paper seems to be correct.
**Clarity:** The paper is very well written.

**Review:**

The evaluation of watermarking approaches is a timely topic and watermarks are one important line of defense against the misuse of generated content. Watermarking has for sure its weaknesses which shows the importance of the following work.

Nevertheless, I have a few concerns that I want to address here:

It would be beneficial to consider a wider range of potential attacks or, at the very least, ensure that the benchmark is designed to be easily expandable to accommodate future attacks.

The evaluated watermarks are confined to post-hoc methods and do not encompass those methods that are inherently integrated into the generation process from the start.

Some additional minor points:

- For the no-box evaluation also results for less than 3.0 for ViQOL could be considered
- In Section 5.2, the following part is a little unclear “First, FNRs of each watermarking method on both datasets are close to 0 for a wide range of detection threshold $\tau$ , indicating that watermarked audios can be accurately detected as watermarked. Second, FPRs of each watermarking method on both datasets decrease as detection threshold $\tau$ increases. This is because unwatermarked audios are less likely to be falsely detected as watermarked when $\tau$ increases”. Are FNR and FPR mixed here?
- What augmentation was used for adversarial training of the tested approaches and are these correlated with the tested ones?

**Strengths:**

- The benchmark is related to a timely topic and the paper well written
- the evaluation is comprehensive and considers various metrics to assess the watermarks

**Additional Feedback:**

No additional feedback

**Documentation:**

The documentation of the benchmark is very limited and more details needs to be added

**Ethics:**

No ethical concerns

**Limitations:**

The limitations are addressed in the paper.

**Opportunities For Improvement:**

- More potential attacks should be considered or at least the benchmark should be easily expandable with future attacks.
- The evaluated watermarks are limited to post-hoc methods and no that are an inherent part of the generation

**Relation To Prior Work:**

The related work is limited and not really addressed.

**Summary And Contributions:**

The paper proposes a benchmark for evaluating audio watermarks against removal attacks. For this, the paper considers different kinds of threat models, including no-box, white-box, and black-box attacks.

---

> ### Author Rebuttal · Authors · 2024-08-19
>
> We appreciate the reviewer's valuable insights and positive score. We've addressed their questions and suggestions in detail below.
>
> ### Comment-1: Detailed documentation.
> ### Response-1:
> Thanks for pointing this out! We are working on expanding the documentation in our GitHub repository to include additional details.
>
> ### Comment-2: ViSQOL scores less than 3.0 in no-box attacks.
> ### Response-2:
> The results for ViSQOL scores below 3.0 in no-attack scenarios are presented in Figures 13 through 16 in the Appendix.
>
> ### Comment-3: Clarification in FNR/FPR in Section 5.2.
> ### Response-3:
> We apologize for the confusion caused by our lack of elaboration. In fact, our original statement is correct. In particular, we consider watermarked audio as positive and unwatermarked audio as negative. Consequently, as the threshold $\tau$ in Figure 2 increases, unwatermarked audio is less likely to be misclassified as watermarked, leading to lower FPRs. We will clarify this further in the revision.
>
> ### Comment-4: Augmentation used in adversarial training
> ### Response-4:
> We summarize the data augmentations and key parameter ranges used in the adversarial training of the three watermarking methods, as well as those applied in our evaluation settings, in Table 8 of the attached PDF. A brief description of each perturbation is provided in Table 2 of Appendix A.2 in our paper. We will add it to our paper during revision.
>
> ### Comment-5: Not considering inherent watermark generation.
> ### Response-5:
> Thanks for pointing this out! This work focuses on post-hoc watermarking methods because they are applicable to any AI-generated audio and are expected to be widely adopted due to their plug-and-play nature. We will clarify this in the introduction and address it further in the future work section.

---

> > ### Comment · Reviewer_1ur8 · 2024-08-20
> >
> > Thank you for the clarifications. I still fully support this work.
> >
> > The augmentation details help to assess the results better, and thank you for pointing to the results regarding less than 3.0 in the Appendix.

---

### Official Review · Reviewer_obkV · 2024-07-19
**The paper presents AudioMarkBench, a benchmark aimed at assessing the robustness of audio watermarking methods against removal and forgery attacks. It evaluates these techniques across different attack scenarios, revealing significant vulnerabilities in current methods.  However, some weaknesses are raised. Firstly, the paper focus mainly on speech domain, overlooking the broader spectrum of AI-generated audio, such as music and sound. And the benchmark is lack in black-box and white-box attack methods. Reviewer requests more experiments to analyze the impact of composite no-box attack and audio’s inherent properties on audio watermarking.**

**Rating:** 4
**Confidence:** 3
**Correctness:** Yes.
**Clarity:** Yes.

**Review:**

This paper is clearly written, has a certain originality and workload, and has played a role in building standardized testing procedures in this field.

Strengths:
1.The paper considers different speaker attributes in the evaluation dataset and analyzes the performance differences of watermarking techniques across various categories.
2.The paper systematically examines the relationship between noise levels and watermark effectiveness, providing a quantitative assessment.

Weakness and Questions:
1.There are many kinds of AI-generated audio, including sound, music, speech, and more. The paper's dataset is mainly focus on speech domain. Although speech is relatively one of the most sensitive audio categories, AI-synthesized audio such as music and sound also pose privacy and infringement risks. Evaluation datasets should cover wider data domains to comprehensively assess the effectiveness of current watermarking techniques and attack methods.
2.The attack methods tested by the paper are not balanced. The paper focus on no-box attacks, but lack in black-box and white-box attacks(e.g., black-box attacks based on gradient estimation). Looking forward to additional supplementary experiments.
3.In practice, audio transmission is likely to undergo multiple compressions, filtering, or resampling processes. Has the evaluation considered the impact of cascading or composite watermark removal and forgery operations?
4.Has this work considered the impact of audio inherent properties, such as the amplitude and frequency distribution of the original audios, on watermarking effectiveness?

**Strengths:**

This work acts as a standardisation benchmark in the audio watermarking field. See the review section for more information.

**Additional Feedback:**

N/A

**Documentation:**

The material provided in the paper illustrate the reproducibility.

**Limitations:**

Yes.

**Opportunities For Improvement:**

See the review section for more information.

**Relation To Prior Work:**

Yes.

**Summary And Contributions:**

The paper present AudioMarkBench, a benchmark designed to evaluate the robustness of audio watermarking methods against both watermark removal and forgery attacks.Moreover, the study benchmarks the robustness of current audio watermarking techniques under several attack scenarios, including no-box, black-box, and white-box settings. The findings highlight significant vulnerabilities in existing methods.

---

> ### Author Rebuttal · Authors · 2024-08-19
>
> Thank you for your hard work and constructive feedback!
>
> ### Comment-1: Wider data domains such as music.
>
> ### Response-1:
> We conducted additional experiments using the FMA music dataset [1]. Experimental results are attached in the pdf file. Specifically, we evaluated the performance of three watermarking methods under various settings:
>
> - **No attacks**: Results are shown in **Table 1**.
> - **No-box attacks**: Results on two representative attacks (background noise and time stretch) are shown in **Table 2**.
> - **Black-box attack**: Results of the Square attack are shown in **Table 3**.
> - **White-box attacks**: Results on the PGD attack are shown in **Tables 4** and **5**.
>
> For the experiments with no attacks and no-box attacks, we randomly sampled 200 unwatermarked audio samples from the FMA validation set. Due to time constraints, we selected 20 of these samples to run experiments on the white-box and black-box attacks. We have several observations from the results:
>
> - **No perturbation**: Under no perturbation, all watermarking methods are accurate across various detection thresholds.
>
> - **No-box attacks and black-box Square attack**: Audioseal and Audioseal-B show better robustness than the other watermarking methods under background noise and black-box Square  attacks. Timbre shows the best robustness under time stretch. These observations are consistent with findings from other datasets presented in our paper.
>
> - **White-box removal and forgery attacks**: All watermarking methods are highly vulnerable to these attacks, with FNRs and FPRs consistently reaching 100% when SNR = 20.
>
>
> ### Comment-2: The attack methods tested by the paper are not balanced.
>
> ### Response-2:
> We appreciate your insightful feedback. In our paper, we evaluated two black-box attacks—HopSkipJumpAttack (HSJA) and Square Attack; as well as white-box watermark-removal and watermark-forgery attacks using Projected Gradient Descent (PGD). In particular, HSJA is a black-box attack based on gradient estimation. Details of these attacks can be found in Section 3 and Appendix A.3 and A.4. Their experimental results are shown in Sections 5.4 and 5.5. In response to your comment, we further added white-box attack experiments using the Iterative-FGSM (I-FGSM) [2]. The results are presented in **Table 6**. We observe that all watermarking methods are highly vulnerable to watermark-removal and watermark-forgery perturbations based on I-FGSM as well, with FNRs and FPRs consistently reaching 100% when SNR = 20. Regarding black-box attacks, we have aimed to incorporate a broad range of state-of-the-art techniques. If you have specific black-box attacks in mind that you would like us to consider, please let us know, and we will be happy to include them in the revision.
>
> ### Comment-3: Composite watermark removal and forgery operations.
> ### Response-3:
> **Table 7** below presents additional results where we explore different combinations of EnCodeC, Gaussian noise, and MP3 compression. We find that EnCodeC remains effective at removing watermarks when MP3 compression is applied, regardless of whether MP3 is applied before or after EnCodeC. Additionally, when Gaussian noise and MP3 compression are combined, the resulting composite attack is shown to be stronger than either component alone, achieving higher FNR and FPR than when Gaussian noise and MP3 are applied independently.
>
> ### Comment-4: Impact of audio inherent properties, such as the amplitude and frequency, on watermarking effectiveness.
> ### Response-4:
> Thank you for your insightful comment. We provide the distribution of the watermarked audio’s average amplitude and the average frequency of the top 10% amplitude after applying a fast Fourier transform (FFT). Results are shown in **Figure 1**. Across all amplitude and frequency ranges shown, all three watermarking methods achieve 0% FNRs, indicating high effectiveness across a broad spectrum of audio properties.
>
> ### Reference
> [1] Defferrard, Michaël, et al. "FMA: A dataset for music analysis." ISMIR, 2017.
>
> [2] Kurakin, Alexey, I. Goodfellow, and Samy Bengio. "Adversarial examples in the physical world. arXiv 2016." ICLR, 2017.

---

> ### Author Response · Authors · 2024-08-29
>
> Dear Reviewer obkV,
>
> With the discussion period ending in two days, we would greatly appreciate any further feedback you may have on our rebuttal. Your insights have been invaluable in refining our work, and we would be glad to answer any additional questions. Thank you once again for your time and effort!
>
> Regards,\
> Authors

---

### Author Rebuttal · Authors · 2024-08-19

We sincerely thank the reviewers for their hard work and constructive feedback! We’ve incorporated substantial experiments as per their request and respectfully request that they consider adjusting their scores if their concerns are addressed by any of the new experiments.

---

### Decision · Program_Chairs · 2024-09-26

**Decision:**

Accept (Poster)

**Comment:**

**Summary**

The paper presents the benchmarking of SOTA watermarking algorithms for the generative speech for a variety of perturbations (no-box, white-box and black-box). The main important outcome of this benchmarking 1) dataset AudioMarkBench inherited from Common Voice which spans over several languages and different speech properties (gender, age, etc); 2) revealing that current audio watermarking algorithms fail to be robust, thus new methods should be develop to provide privacy and safety.

**Summary of the issues / contributions**

All reviewers agreed that the paper is well written, is an important topic, has extensive experiments and comprehensive evaluation, has valuable observation that current watermarking is  vulnerable and has significant gender bias, considers different speaker attributes in the analysis.  **Reviewer 1ur8** pointed out that “The evaluation of watermarking approaches is a timely topic and watermarks are one important line of defense against the misuse of generated content. Watermarking has for sure its weaknesses which shows the importance of the following work.”

Several issues listed by **Reviewer M2hh** (e.g. limitation of the dataset, number of SOTA models) and **Reviewer 1ur8** (focus is only on post-hoc watermarking) are responded and resolved in the rebuttal. **Reviewer obkV** pointed out the absence of analysis of watermarking for music generation (authors provided such evaluation on one dataset), absence of ablations on different aspects (authors added comprehensive ablations and results are consistent with the stated observations in the main paper) and diversity of attacks (authors added one more white-box attack and pointed out all attacks they used). However, **Reviewer obkV** did not participate in the discussion period and did not acknowledge the authors rebuttal — thus I ignore the rating from the reviewer as authors responses in the rebuttal are valid and comprehensive in my opinion.

**Recommendation**

2/3 reviewers are in acceptance of the work.  The rating from the third reviewer is ignored in the final decision making based on the above points.

As generative models involve past years and speech is a sensitive domain, I believe authors consider an important benchmark and I agree with reviewers pointing out the high demand of this evaluation now. Based on the importance of the work for privacy, fairness and safety issues of speech generative models as well as comprehensive analysis and valuable findings, **I recommend the paper for acceptance**. I encourage authors to include all new results provided during rebuttal, responses to reviewers as well as instructions/discussion how the benchmark can be expanded to other watermarking methods.